# An engineered ligand trap inhibits leukemia inhibitory factor as pancreatic cancer treatment strategy

Sean A. Hunter [1], Brianna J. McIntosh [1], Yu Shi[2], R. Andres Parra Sperberg[3], Chie Funatogawa[4], Louai Labanieh[3], Erin Soon[5], Hannah C. Wastyk[3], Nishant Mehta[3], Catherine Carter[3], Tony Hunter[2] & Jennifer R. Cochran [1,3,5,6 ✉]

Leukemia inhibitory factor (LIF), a cytokine secreted by stromal myofibroblasts and tumor cells, has recently been highlighted to promote tumor progression in pancreatic and other cancers through KRAS-driven cell signaling. We engineered a high affinity soluble human LIF receptor (LIFR) decoy that sequesters human LIF and inhibits its signaling as a therapeutic strategy. This engineered 'ligand trap', fused to an antibody Fc-domain, has ~50-fold increased affinity (~20 pM) and improved LIF inhibition compared to wild-type LIFR-Fc, potently blocks LIF-mediated effects in pancreatic cancer cells, and slows the growth of pancreatic cancer xenograft tumors. These results, and the lack of apparent toxicity observed in animal models, further highlights ligand traps as a promising therapeutic strategy for cancer treatment.

[1] Cancer Biology Program, Stanford University School of Medicine, Stanford, CA, USA. [2] Molecular and Cell Biology Laboratory, Salk Institute for Biological Studies, La Jolla, CA, USA. [3] Department of Bioengineering, Stanford University, Stanford, CA, USA. [4] Unchained Labs, Pleasanton, CA, USA. [5] Immunology Program, Stanford University School of Medicine, Stanford, CA, USA. [6] Department of Chemical Engineering, Stanford University, Stanford, CA, USA. ✉email: jennifer.cochran@stanford.edu

Pancreatic ductal adenocarcinoma (PDAC) is the fourth leading cause of cancer-related deaths in the United States, with a grim 5-year survival rate of only 9%[1]. The intractability of PDAC to treatment is due to multiple causes, notably late diagnosis of disease, dense stroma surrounding the tumor making drug delivery challenging, and a general dearth of effective therapies[2]. Recently, leukemia inhibitory factor (LIF) has been identified as a promising target for therapeutic intervention[3–8]. LIF is a pleiotropic cytokine member of the interleukin-6 (IL-6) family that signals through a heterodimeric complex consisting of LIF receptor (LIFR) and glycoprotein 130 (gp130) proteins (Supplementary Fig. 1a)[9]. LIF-mediated receptor dimerization activates the Janus kinase family and downstream signaling through signal transducer and activator of transcription 3 (STAT3), phosphatidylinositol 3-kinase, and the ERK mitogen-activated kinase pathway, among other effectors (Supplementary Fig. 1a). LIF has been shown to be produced at high levels in a variety of tumor types and promotes proliferation, invasiveness, stemness, neuronal remodeling, and the epithelial-to-mesenchymal transition[3–8,10–16].

Recent studies have identified LIF expression in the tumor microenvironment as a major driver of PDAC progression[3–7]. Notably, LIF production is upregulated via prolonged KRAS signaling, a hallmark of PDAC[2,4,17,18]. LIF is not only secreted at high levels by PDAC cells, but also from the associated myofibroblastic stellate cells in response to signals from PDAC cells (Supplementary Fig. 1a)[3]. The resultant effect of these elevated LIF levels is multifaceted. In PDAC cells, LIF is the major cytokine responsible for STAT3 activation in tumor cells, a critical factor in PDAC initiation, progression, and maintenance[3,19,20]. LIF is also a known stem-cell factor, and LIF inhibition has been shown to ablate the ability of PDAC cells to initiate tumors[4,21,22]. Further, LIF blockade causes existing PDAC tumors to become less aggressive and more differentiated, likely due to loss of cancer stem cells[3]. Taken together these results identify LIF as a novel, complex therapeutic target in PDAC.

Conditional genetic deletion of LIFR, the major LIF receptor, or therapeutic inhibition of LIF, has led to prolonged survival in mouse models of PDAC[3,4]. Genetic knockout of both copies of *LIFR* leads to significantly longer survival in a mouse model of pancreatic cancer[3] that mimics human disease progression (KP$^{f/f}$CL model: Kras$^{LSL-G12D/+}$;Trp53$^{flox/flox}$;Pdx1-Cre;Rosa26$^{LSL-Luc/LSL-Luc}$)[23,24]. In addition, a significant increase in lifespan and decrease in tumor mass has been observed in KP$^{f/f}$CL mice upon treatment with an anti-LIF monoclonal antibody in combination with chemotherapy[3]. Antibody-based anti-LIF therapy was also effective in slowing tumor growth in cell line and patient-derived xenograft models of human PDAC[4]. These results highlight the importance of LIF in driving PDAC, demonstrate tumor susceptibility to LIF inhibition, and highlight an opportunity for effective LIF inhibitors, none of which are yet clinically approved.

Here, we engineered a soluble, recombinant variant of the human LIFR extracellular domain that acts as a 'ligand trap' by binding to and sequestering human LIF from its activating receptors (Fig. 1a). LIFR variants with increased binding affinity for LIF were engineered and isolated using combinatorial protein engineering via yeast-surface display[25,26]. We fused the highest affinity variant to an antibody Fc domain to create a construct that binds LIF with 50-fold higher apparent affinity ($K_d \sim 20$ pM) as compared to wild-type (WT) LIFR and competes directly with LIFR for binding. This engineered protein potently inhibits LIF-derived signaling and sphere formation in PDAC cells and slows tumor progression in a mouse model of disease.

## Results

### Treatment with an mLIFR-Fc ligand trap is non-toxic and blocks LIF signaling in vivo. Based on the therapeutic efficacy of

ligand traps in treating other forms of cancer[27–29], we first sought to interrogate the therapeutic potential of a LIFR-based ligand trap in the KP$^{f/f}$CL genetic model of PDAC by designing an inhibitor of mouse LIF (mLIF), based on mouse LIFR (mLIFR). We generated this surrogate because while mLIFR can bind human LIF, human LIFR (hLIFR) cannot effectively bind to murine LIF. Soluble mLIFR composed of the three most N-terminal extracellular domains, cytokine-binding motif I (CBM I)–Ig-like–cytokine-binding motif II (CBM II), has previously been shown to inhibit mLIF derived signaling[30]. We fused the sequence of mLIFR to the fragment crystallizable (Fc) domain of an antibody in order to create an Fc-fusion protein (mLIFR-Fc), which has benefits including extended serum half-life, bivalency, and Fc-receptor recycling[31,32]. To assess whether a ligand trap was a viable therapeutic approach, we first showed that soluble mLIFR-Fc inhibited mLIF induced phospho-STAT3 (pSTAT3) in a PDAC cell line derived from the pancreas of a KP$^{f/f}$CL mouse (Fig. 1b, Supplementary Fig. 1b). The mLIFR-Fc exhibited reduced efficacy and bound to yeast-displayed mLIF with five-fold weaker affinity than the D25[33] anti-LIF monoclonal antibody (mAb) (apparent $K_d$ values of $160 \pm 20$ pM versus $33 \pm 8$ pM, respectively) (Supplementary Fig. 1c).

We next determined whether mLIFR-Fc could block LIF activity in vivo in the KP$^{f/f}$CL PDAC genetic mouse model (Fig. 1c). KP$^{f/f}$CL PDAC replicates human disease[23,24], including the presence of a dense stroma surrounding the tumor, which can serve as a physical barrier for effective therapeutic delivery[34]. We observed robust pSTAT3 inhibition in cancerous regions of the pancreases of KP$^{f/f}$CL mice upon treatment with mLIFR-Fc, to a degree comparable with or greater than the D25 antibody (Fig. 1d). These results provide proof-of-concept evidence that mLIFR-Fc can block the effects of LIF activity on tumors.

Potential toxicity is a critical consideration of any therapeutic agent. LIF inhibition is expected to be well tolerated, as LIF expression is relatively low in adult mice and humans, while mutations or deletions of the LIF gene do not lead to negative health consequences, save for women presenting lower conception rates likely due to the fact that LIF is required for embryo implantation[3,35,36]. Further, no toxicity was observed in mice treated with anti-LIF antibodies[3,4]. We sought to determine the effects of treatment with a LIFR-based ligand trap, dosing two strains of non-tumor-bearing WT mice (FVB and Black/6) three times per week with phosphate-buffered saline (PBS) or 20 mg/kg mLIFR-Fc for 1 month. Over the course of the treatment, neither strain lost weight (Fig. 1e), and neither showed any other visible sign of toxicity, such as hunching or matted fur. After 1 month, blood samples and organs were collected and analyzed by an independent pathologist. We observed no consistent changes in blood analytes between strains when comparing mLIFR-Fc and saline treated mice (Supplementary Table 1). Further, none of the 29 organs examined by an independent pathologist, including the heart, kidney, liver, and pancreas, showed signs of toxicity (Fig. 1f). Together, these results demonstrate that a LIFR-based ligand trap can effectively block LIF-mediated LIFR signaling in PDAC, without evidence of toxicity.

### Engineering hLIFR to bind with increased affinity to LIF. We next created a ligand trap based on human LIFR (hLIFR), as a mouse-derived protein is an unsuitable therapeutic due to a high likelihood of immunogenicity. We used combinatorial methods to engineer hLIFR variants with greatly increased affinity to human LIF as our previous work has shown that higher affinity ligand traps are more potent inhibitors[27–29]. Of hLIFR's six extracellular domains (Supplementary Fig. 2a), only the three

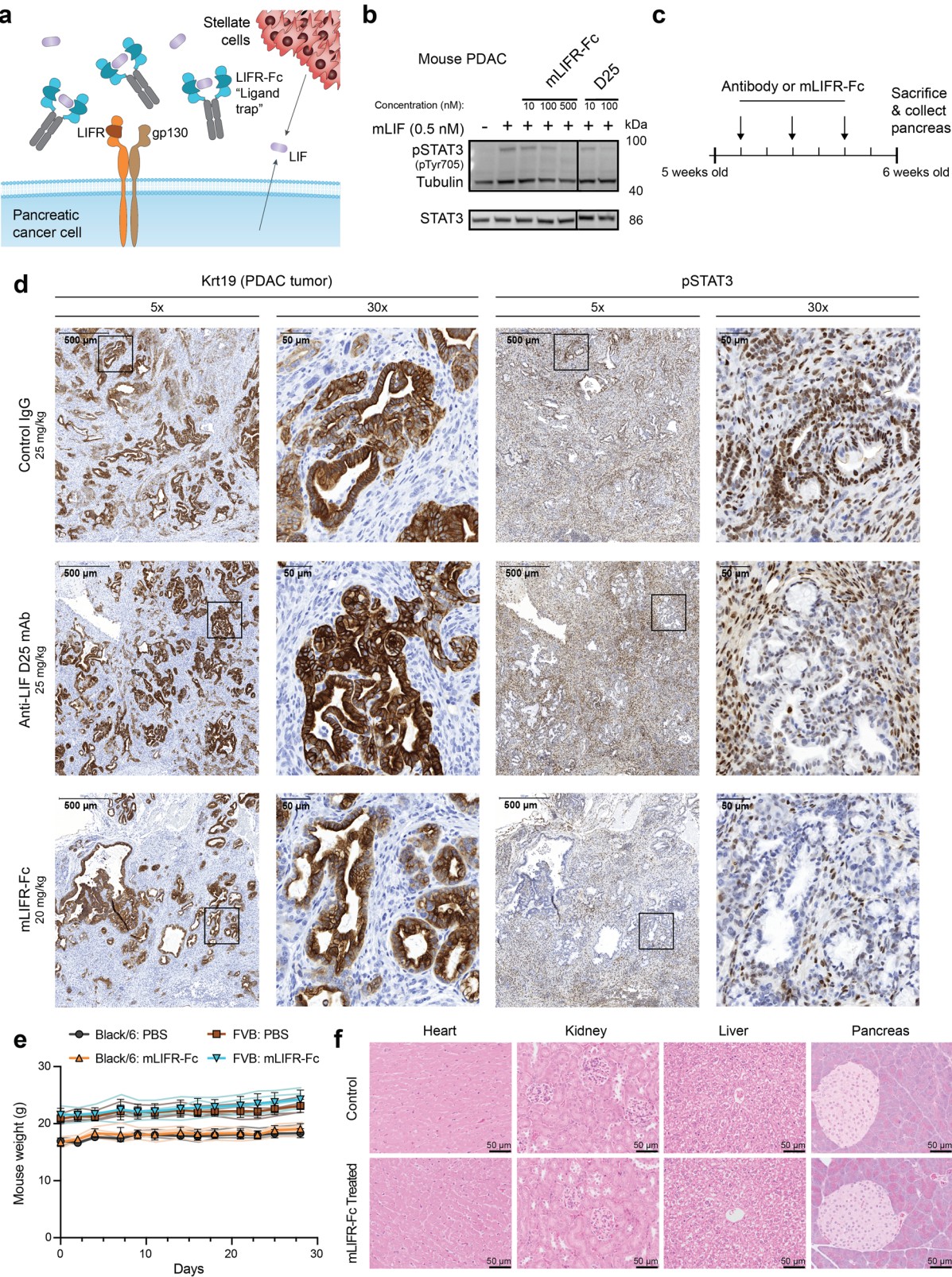

most N-terminal domains contribute to LIF binding: CBM I, Ig-like, and CBM II, with most of the LIF–LIFR contacts mediated by the Ig-like domain[9,37–39]. We displayed various combinations of these three domains on the yeast cell surface and used flow cytometry to measure functional protein expression via antibody detection of the C-terminal c-myc tag, and cytokine binding via antibody detection of histidine-tagged human LIF (LIF-His)

(Fig. 2a). High levels of expression (Supplementary Fig. 2b) and LIF binding (Supplementary Fig. 2c) were observed with the induced display of two domains (CBM I and Ig-like domains), or all three domains (CBM I, Ig-like, and CBM II domains), but no binding occurred when the three domains were displayed individually (Supplementary Fig. 2c). We moved forward with the CBM I–Ig-like domain combination as a starting point for

**Fig. 1 mLIFR-Fc inhibits LIF signaling in PDAC in KP$^{f/f}$CL mice and is non-toxic. a** The LIFR extracellular domain, repurposed as a soluble "ligand trap", binds to LIF in the tumor microenvironment and blocks LIF signaling. **b** Immunoblotting of lysates from mouse PDAC cells treated as indicated. Lanes were taken from the same blot, with solid lines indicating lanes that were not adjacent. Uncropped blots and gels from this and subsequent blots and gels can be found in Supplementary Fig. 8. **c** Dosing schedule (arrows) for test of pSTAT3 reduction in KP$^{f/f}$CL mice. **d** mLIFR-Fc robustly inhibits pSTAT3 signaling in PDAC tumors. Representative pancreas sections from KP$^{f/f}$CL mice treated with a control IgG (25 mg/kg, 3 times), the D25 mAb (25 mg/kg, 3 times), or mLIFR-Fc (20 mg/kg, 3 times) stained via immunohistochemistry for Krt19 (PDAC lesions) and pSTAT3. Two different magnifications (×5 and ×30) are shown, with zoomed in regions identified by boxes ($n = 3$ mice/group). **e** FVB (PBS: brown squares; mLIFR-Fc: blue inverted triangles) and Black/6 (PBS: gray circles; mLIFR-Fc: orange triangles) mice did not experience weight loss when injected with mLIFR-Fc (20 mg/kg) or phosphate-buffered saline (PBS; equal volume) 3×/week, intraperitoneally (IP). Mice were weighed at each injection. Data represent the mean ± SD, with data from individual mice represented as faint lines ($n = 5$ mice/group). **f** No toxicity was evident via representative hematoxylin and eosin staining of organ slices from PBS-injected and mLIFR-Fc-injected FVB mice.

engineering studies, as it was the minimal set of extracellular domains that conferred LIF binding.

A library of LIFR variants was generated by error-prone PCR, targeted to the Ig-like domain to maximize discovery of beneficial mutations across a smaller sequence space. We transformed mutated DNA into the yeast strain EBY100, resulting in a library of ~$2.7 \times 10^7$ transformants of LIFR variants, individually expressed on the yeast cell surface. The library was screened by fluorescence-activated cell sorting (FACS) to isolate the top binders to LIF-His over six increasingly stringent rounds of sorting (Fig. 2b, Supplementary Fig. 2d). The yeast pool remaining after the final sort bound to 100 pM LIF more strongly than yeast-displayed WT hLIFR (Supplementary Fig. 2e), as measured by flow cytometry. DNA sequencing of yeast plasmids isolated from sorts 4 through 6 revealed clear mutational consensus at positions L218P and N277D, and partial consensus at N242D, I257V, and V262A (Fig. 2c). We confirmed that the most prevalent variant L218P–N242D–I257V–N277D ('PDVD'; Fig. 2c) had greatly improved binding to LIF over hLIFR at 100 pM LIF (Supplementary Fig. 2f). Mutation N277D was previously reported to improve the affinity of hLIFR to LIF, validating the method and approach of our screening strategy[38].

To further improve LIF-binding affinity, we created a second-generation LIFR library, using error-prone PCR for additional mutagenesis and DNA shuffling to recombine mutations. We again transformed mutated DNA into yeast to generate a LIFR-displayed library of ~$1.6 \times 10^8$ transformants, which we screened for top binders to LIF-His over five increasingly stringent rounds of sorting (Supplementary Fig. 2g). The final yeast pool bound the very low concentration of 10 pM LIF much more strongly compared to yeast-displayed hLIFR, for which binding was barely detectable (Supplementary Fig. 2h). DNA sequencing of yeast plasmids isolated from sorts 4 and 5 identified five additional consensus mutations in addition to the previously identified mutations: I217V, H240R, N242S, I260V, and T273I (Fig. 2d).

To parse contributions of consensus mutations, we used site-directed mutagenesis to create variants that contained combinations of mutations that did not occur naturally in the library (Fig. 2e). We then compared these variants to WT hLIFR, and the most common variants from library 1 ('PDVD') and library 2 ('PRVAID' and 'PRDSAID'), measuring a binding score reflective of fold improvement over wild type (Fig. 2e). Combining mutations generally improved binding, and a final variant containing eight mutations (eLIFR: I217V, L218P, H240R, I257V, I260V, V262A, T273I, and N277D; 'VPRVVAID') had the highest relative affinity (Fig. 2e). Many of these mutations were located within or adjacent to the proposed LIF–LIFR binding interface in a computational model of the ligand-receptor co-complex structure (Fig. 2f). Yeast-displayed eLIFR generated a significantly higher binding signal than WT hLIFR (Fig. 2g), with an apparent affinity for eLIFR containing the CBM I and Ig-like domains of 40 ± 6 pM (Fig. 2g).

**eLIFR-Fc is a potent inhibitor of LIF.** Next, hLIFR and eLIFR were genetically fused to the fragment crystallizable (Fc) domain of mouse IgG2a or human IgG1 antibodies in a variety of orientations to create Fc-fusion proteins (Supplementary Fig. 3a, b). In these constructs, we included the entirety of all three domains (CBM I–Ig-like–CBM II), exploiting increased affinity from interactions of LIF with the CBM II domain (Fig. 3a). As recombinant proteins, an N-terminal eLIFR-Fc fusion and C-terminal eLIFR-Fc fusion containing a (Glycine)$_4$Serine linker had the strongest affinities for yeast-surface displayed LIF (apparent $K_d$ values of 10 pM and 20 pM, respectively), likely from avidity effects of these bivalent constructs compared to a monovalent eLIFR-Fc fusion (apparent $K_d = 130$ pM) (Supplementary Fig. 3c). An anti-LIF mAb, D25[3,4,33], had an apparent $K_d$ value of 70 pM. We brought forward the N-terminal eLIFR-Fc fusion (henceforth referred to as "eLIFR-Fc"; Fig. 3b) for further studies, as this configuration mimics the natural orientation of the Fab arms to the Fc domain in antibodies. Purified eLIFR-Fc and hLIFR-Fc were assessed by analytical size exclusion chromatography (Supplementary Fig. 3d). Both purified proteins displayed near identical melting temperatures and a strikingly limited propensity for aggregation, especially when compared to a control IgG, as measured using intrinsic fluorescence, static light scattering (SLS), and dynamic light scattering (DLS) on an Unchained Labs Uncle instrument (Supplementary Fig. 3e).

As a further measure of increased affinity, we showed that eLIFR-Fc maintained significantly higher binding to yeast-displayed LIF upon extended washing and incubation with high concentrations of soluble LIF competitor, suggesting a slower off-rate compared to hLIFR-Fc (Fig. 3c). We next measured the affinity of eLIFR-Fc and hLIFR-Fc using the Kinetic Exclusion Assay (KinExA), where both the Fc-fusions and LIF were in soluble form. From this assay, the apparent $K_d$ of eLIFR-Fc was measured to be 23 ± 9 pM, an approximately 50-fold improvement in affinity over that of hLIFR-Fc, which was 1.1 ± 0.4 nM (Fig. 3d). We also tested the ability of the proteins to inhibit LIF binding to yeast-displayed WT LIFR or gp130. Interestingly, the D25 antibody out-competed LIF binding to LIFR (Fig. 3e) but did not compete with LIF in binding to gp130 (Fig. 3f). Conversely, hLIFR-Fc and eLIFR-Fc competed LIF from binding LIFR and gp130, with eLIFR-Fc a significantly more potent competitor for both receptors (Fig. 3e,f).

**eLIFR-Fc effectively blocks LIF-derived programming in cancer cells.** We next characterized how well eLIFR-Fc functionally blocks LIF activity using a HeLa reporter cell line that contains the firefly luciferase gene downstream of a triple STAT3 response element (Supplementary Fig. 4a). These cells were responsive over a wide range of LIF concentrations (down to 3 pM LIF), with peak activity at ~1 nM LIF (Supplementary Fig. 4b). Even at LIFR-Fc concentrations of only tenfold excess of LIF, eLIFR-Fc almost completely silenced LIF signaling,

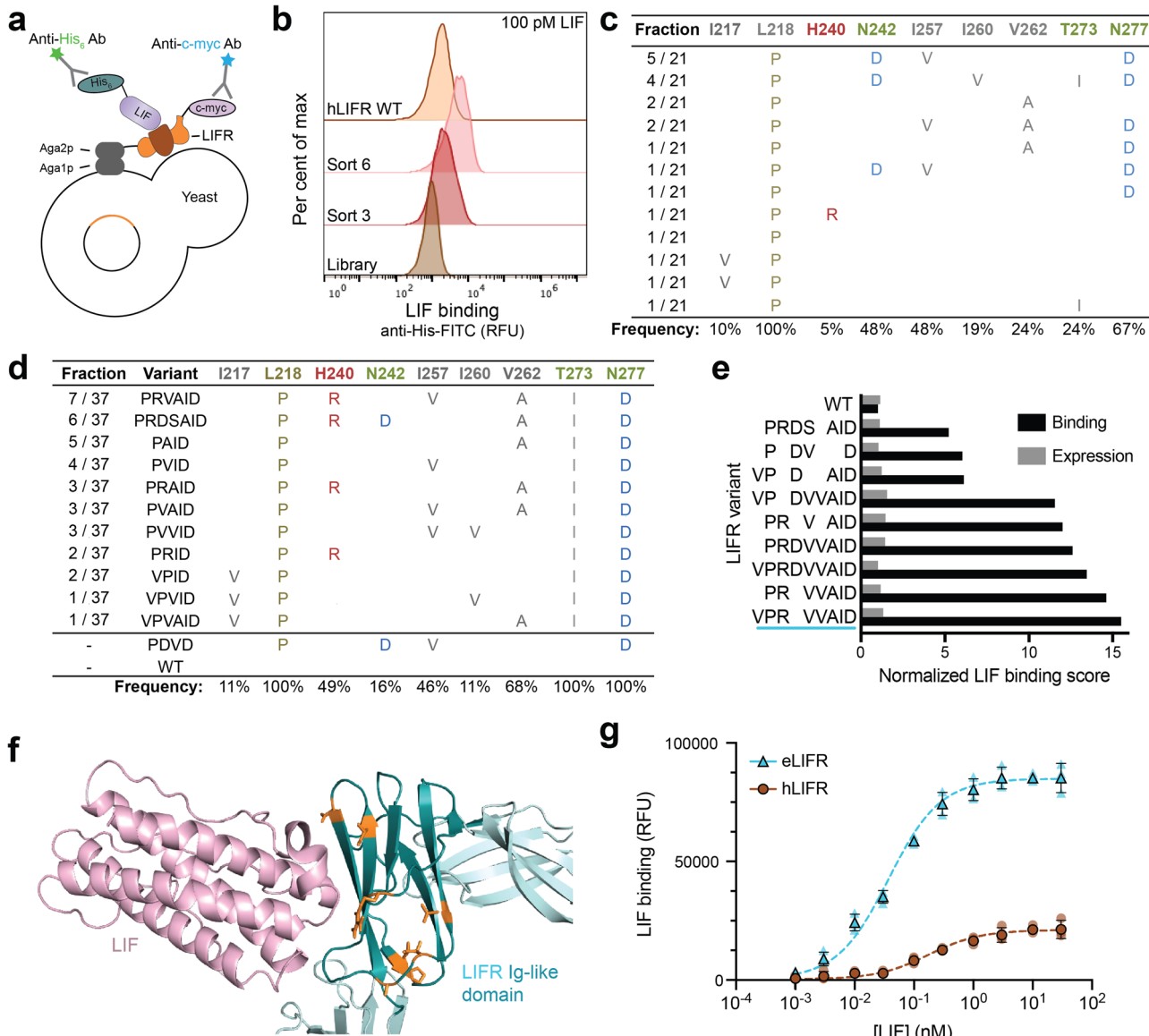

**Fig. 2 Engineering and identification of LIFR variants with increased LIF-binding affinity. a** Cartoon of detection of yeast-displayed LIFR expression and binding to His₆-tagged LIF by flow cytometry. **b** LIFR binding at 100 pM LIF improved over successive rounds of library sorting. Flow cytometry histograms of the unsorted, first-generation LIFR library ("Library") and intermediate sorted populations (sort 3 and sort 6) compared to WT hLIFR. Only the gated population of yeast displaying LIFR are shown. The experiment was repeated at least three times independently with similar results. **c** Consensus mutations from Library 1, identified after six rounds of sorting. Amino acids with similar properties are grouped by color. **d** Unique sequences and consensus mutations from Library 2, sorts 4 and 5, listed in order of frequency. **e** LIF-binding scores of yeast-displayed WT and LIFR mutant variants, derived using site-directed mutagenesis, measured by flow cytometry. **f** Proposed structure of eLIFR in complex with LIF, modeled by inserting mutations into hLIFR WT PDB (3E0G) using Rosetta Remodel ("Methods"). hLIF was docked locally to the eLIFR model using RosettaDock. "VPRVVAID" mutations are shown in orange. **g** Yeast-displayed eLIFR (blue triangles) binds LIF-His with a higher affinity than WT hLIFR (brown circles). Higher maximum binding signal from eLIFR likely indicates a slower off-rate of binding. Binding curves were fit using non-linear regression. Data are the geometric mean of fluorescence of the expressing population ± standard deviation (SD) of triplicate measurements. Data from individual experiments are shown as faint symbols.

while hLIFR-Fc only reduced LIF-driven signaling by ~50% (Supplementary Fig. 4c,d). In these assays, eLIFR-Fc had an IC₅₀ of 35 ± 5 pM, ~50-fold more potent than that of hLIFR-Fc (1.8 ± 0.3 nM; Fig. 4a), consistent with the affinity ranges that were measured for each molecule (Fig. 3d). Since LIF is present at high levels in the PDAC tumor microenvironment[3], we sought to determine the efficacy of each inhibitor added after LIF had already begun to drive signaling. Upon delayed addition, eLIFR-Fc potently silenced LIF activity, while hLIFR-Fc was ineffective, even when added in great excess (Fig. 4b, Supplementary Fig. 4e).

We tested the ability of eLIFR-Fc to directly inhibit LIF-derived signaling via phospho-STAT3 (pSTAT3) in PDAC cells. Upon LIF addition to the media, LIF binds to LIFR and gp130 and initiates a signaling cascade that leads to elevated levels of pSTAT3, with peak signal lasting up to 20 min after cytokine addition (Supplementary Fig. 5a). We found that eLIFR-Fc substantially blocked LIF-derived pSTAT3 signal in both KP4 and PANC1 human PDAC cell lines when added at 7.5-fold excess of LIF (Supplementary Fig. 5b, c). This blockade was mediated through LIFR, as shRNA knockdown of LIFR (LIFR KD) in KP4 and PANC1 cell lines resulted in an absence of LIF

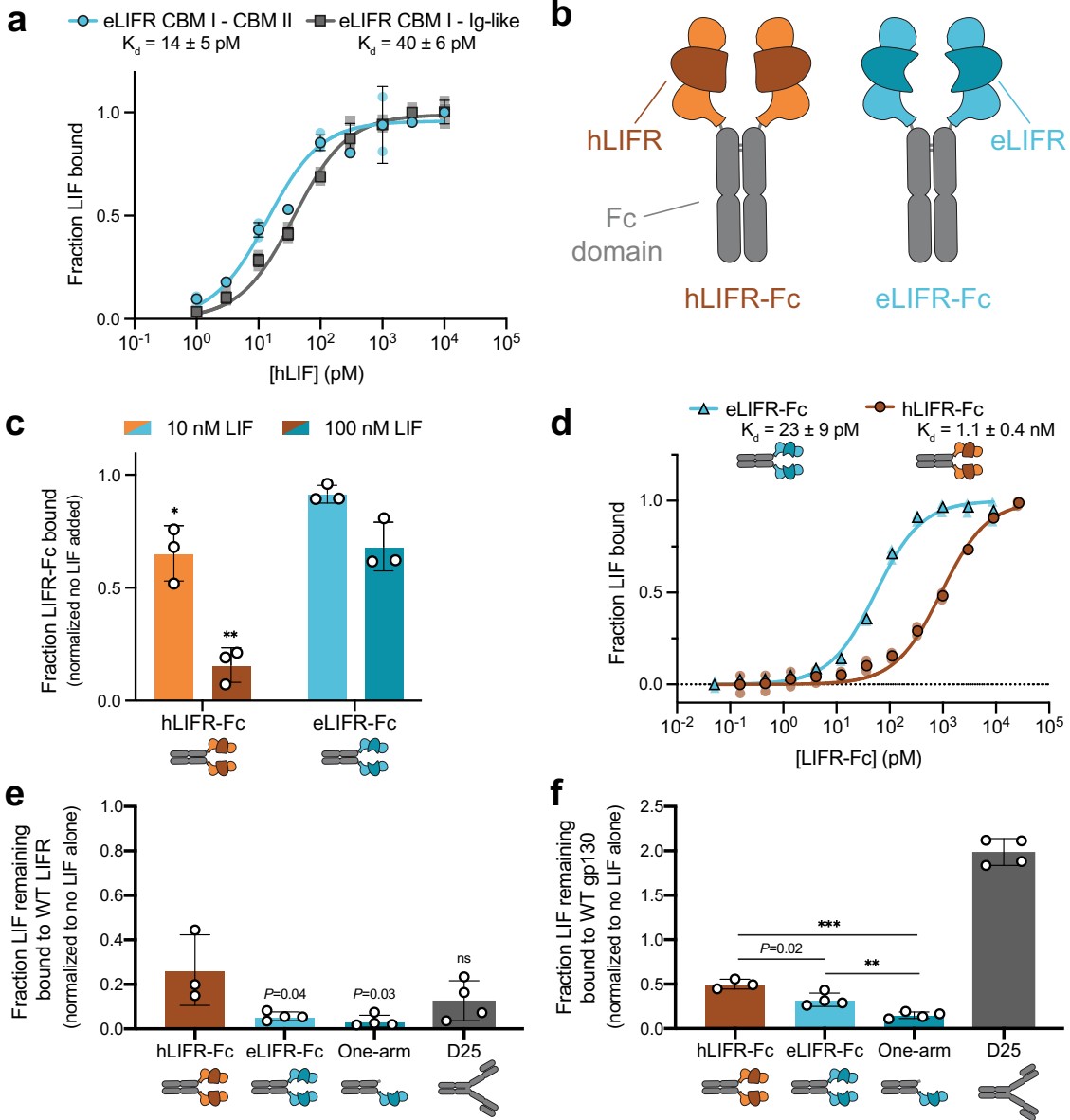

**Fig. 3 eLIFR-Fc is a more potent LIF inhibitor than hLIFR-Fc. a** Yeast-displayed eLIFR containing the CBM I–Ig-like–CBM II domains (blue circles) has a slightly higher affinity for LIF-His than eLIFR CBM I–Ig-like domains (gray squares). Data are the mean ± SD of triplicate measurements. Data from individual experiments are shown as faint symbols. **b** Schematic of hLIFR-Fc and eLIFR-Fc. The three N-terminal domains, CBM I–Ig-like–CBM II, are fused to an hIgG1 Fc-domain, with the engineered Ig-like domain shown in dark teal for eLIFR-Fc. **c** Versus hLIFR-Fc, eLIFR-Fc remains more strongly bound to yeast-displayed LIF after overnight incubation with soluble LIF competitor, indicating a slower off-rate. *$P = 0.024$, **$P = 0.0024$ versus the corresponding hLIFR-Fc condition, two-tailed unpaired Student's $t$ test. Data are mean ± SD ($n = 3$). **d** KinExA data showing that recombinant eLIFR-Fc (blue triangles) has higher affinity to soluble hLIF-His than hLIFR-Fc (brown circles). Data are the mean of duplicate measurements. Data from individual experiments are shown as faint symbols. **e** hLIFR-Fc, eLIFR-Fc (bivalent and one-arm), and D25 antibody compete LIF away from WT LIFR. ns not significant, $P = 0.03$ for one-arm eLIFR-Fc and $P = 0.04$ for eLIFR-Fc versus hLIFR-Fc, two-tailed unpaired Student's $t$ test. **f** Both hLIFR-Fc and eLIFR-Fc (bivalent and one-arm) compete LIF away from WT gp130, but the D25 mAb does not and appears to increase binding, perhaps due to complex stabilization or more avid LIF binding. $P = 0.02$, **$P = 0.006$, or ***$P \le 0.0002$ versus hLIFR-Fc or eLIFR-Fc (as indicated), two-tailed unpaired Student's $t$ test. For **e** and **f**, data are mean ± SD ($n \ge 3$ independent experiments).

signaling (Supplementary Fig. 5b, c). eLIFR-Fc achieved a 50% reduction in signal at only a fourfold excess in concentration over LIF, and completely extinguished LIF signaling at higher concentrations (Fig. 4c, Supplementary Fig. 5d–f). Conversely, treatment with hLIFR-Fc was not able to achieve even a 50% reduction of pSTAT3 levels in KP4 or PANC1 cells, even at concentrations 100-fold in excess of LIF (Fig. 4c, Supplementary Fig. 5d–f). In addition, eLIFR-Fc inhibition compared

favorably to and in some cases out-performed the D25 antibody (Supplementary Fig. 5d, f).

One of the best studied properties of LIF is its role as a pro-survival, stem-cell factor that drives anchorage-independent growth and the formation of cancer spheroids[3,4]. We established KP4 spheroids by adding growth factors FGF and B27 and measuring overall sphere area over time (Supplementary Fig. 5g). LIF addition significantly increased sphere formation, leading to

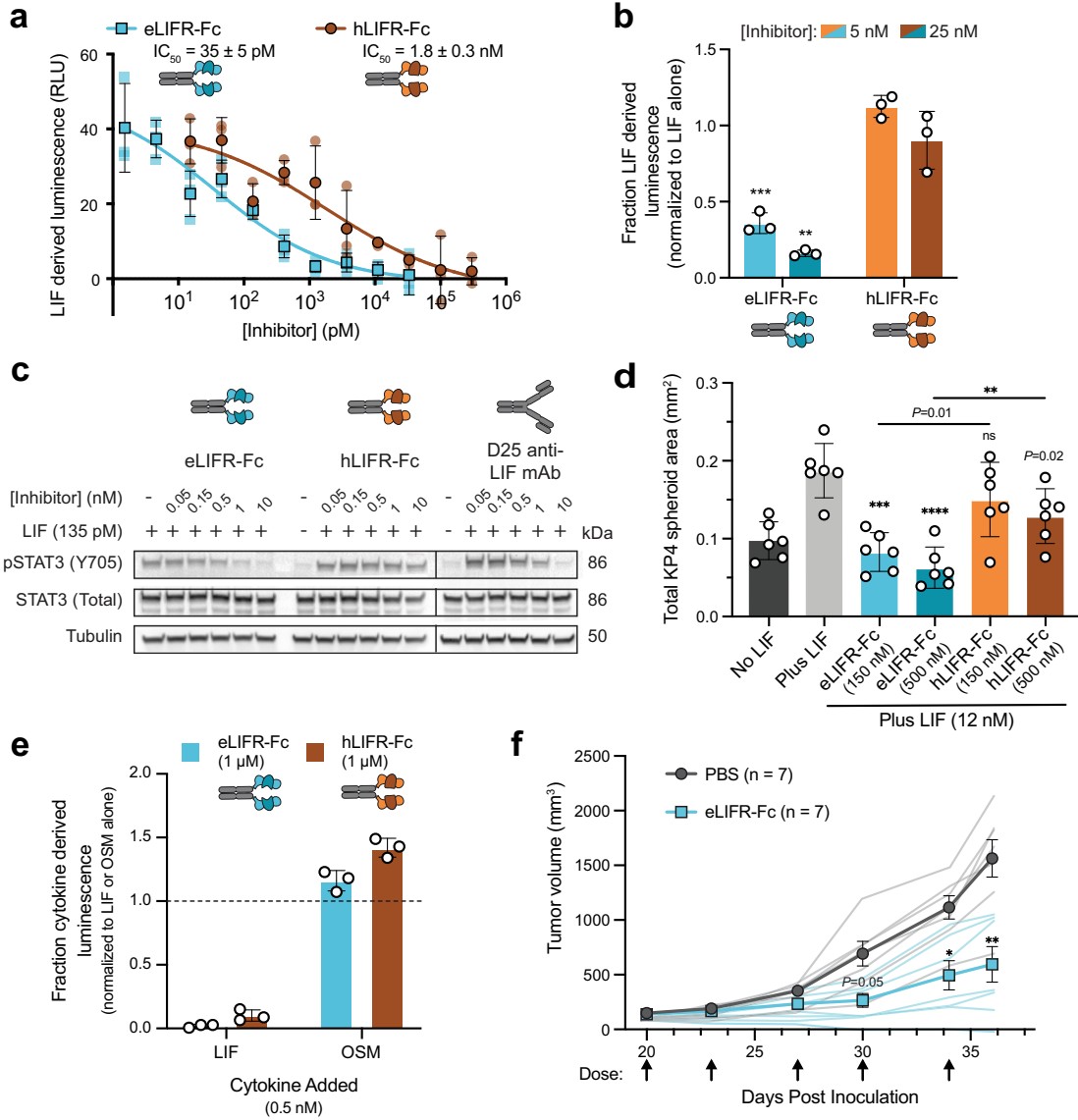

**Fig. 4 eLIFR-Fc inhibits LIF activity in models of cancer. a** eLIFR-Fc (blue squares) functionally inhibits LIF activity with a ~50-fold better IC$_{50}$ than hLIFR-Fc (brown circles) in HeLa STAT3 luciferase reporter cells. [LIF] = 20 pM. Data are the mean ± SD ($n = 3$). Data from individual experiments are shown as faint symbols. **b** Normalized luminescence from HeLa STAT3 luciferase reporter cells incubated with 0.5 nM LIF for 1 h, followed by addition of eLIFR-Fc or hLIFR-Fc as indicated. **$P = 0.0025$, ***$P = 0.0002$ compared to the corresponding hLIFR-Fc condition by two-tailed unpaired Student's *t* test. Data are the mean ± SD ($n = 3$). **c** Immunoblot of cell lysates from KP4 human PDAC cells incubated with LIF and/or each protein as indicated ($n = 1$). Lines indicate lanes that were not adjacent on the original blot. **d** Total spheroid area of KP4 human PDAC cells, 1 week after seeding. Each data point represents an individual well of a 96-well plate; spheres were identified manually and the total area of all spheres in each well was quantified with ImageJ. **$P = 0.0042$, ***$P = 0.0001$, ****$P < 0.0001$ compared to "Plus LIF" condition (for stars and *P* values above bars) or between hLIFR-Fc and eLIFR-Fc (indicated by horizontal lines) by two-tailed unpaired Student's *t* test. Data are the mean ± SD ($n = 6$). **e** High concentrations of eLIFR-Fc or hLIFR-Fc do not inhibit OSM signaling in HeLa STAT3 cells as compared to LIF. Raw luminescence, minus baseline luminescence, was normalized to the luminescence from the respective condition without inhibitor (dashed line). Data are the mean ± SD ($n = 3$ for all except 3 nM LIF, where $n = 2$). **f** Treatment of human KP4 PDAC xenograft model with PBS (gray circles) or eLIFR-Fc (blue squares; 10 mg/kg, 2×/week). Dosing schedule indicated by arrows. $P = 0.05$; *$P = 0.02$; **$P = 0.009$ by two-way ANOVA with Sidak's multiple comparisons test. Data are the mean ± SEM ($n = 7$). Data from individual mice are shown as faint lines.

>2-fold greater aggregate spheroid area (Fig. 4d, Supplementary Fig. 5h). eLIFR-Fc mitigated LIF-driven spheroid formation, even at concentrations only twofold in excess of LIF (Supplementary Fig. 5h), and significantly better than hLIFR-Fc (Fig. 4d, Supplementary Fig. 5h). eLIFR-Fc also significantly inhibited sphere formation by PANC1 cells, whereas hLIFR-Fc was ineffective (Supplementary Fig. 5i). Taken together, these data collectively establish that eLIFR-Fc is a potent functional inhibitor of LIF activity in PDAC cells.

**eLIFR-Fc does not effectively bind to other IL-6 family cytokines.** Other IL-6 family member cytokines share LIFR and gp130 co-receptors including oncostatin M (OSM), cardiotrophin-1 (CTF-1), and cardiotrophin-like cytokine factor 1 (CLCF-1) (Supplementary Fig. 2a)[40,41]. Flow cytometry analysis showed that yeast-displayed CBM I–Ig-like–CBM II domains of hLIFR or eLIFR bind these cytokines very weakly, especially when compared to LIF (Supplementary Fig. 6a). Affinity maturation of receptors toward a specific ligand tends to improve or retain

specificity for that ligand, and the increased binding of eLIFR to LIF was consistent with this trend (Supplementary Fig. 6a)[42].

We next determined the specificity of our constructs in blocking the signaling of LIF versus other IL-6 family members. We detected minimal inhibition of pSTAT3 in KP4 PDAC cells upon co-incubation of eLIFR-Fc with OSM, in stark contrast to complete inhibition upon incubation with LIF (Supplementary Fig. 6b), consistent with the weak OSM binding signal (Supplementary Fig. 6a). HeLa STAT3 luciferase reporter cells generate luminescence signal from all IL-6 family member cytokines, albeit at different potencies (Supplementary Fig. 6c). We observed no significant inhibition of luciferase signal derived from OSM, CTF-1, or CLCF-1, while LIF-derived signal was robustly silenced at high LIFR-Fc concentrations (1 μM) (Fig. 4e, Supplementary Fig. 6d). Unlike LIF, which engages only with LIFR and gp130, the ligands OSM, CTF-1, and CLCF-1 also utilize additional or alternative receptor complexes to drive high-affinity interactions and subsequent cell signaling;[40,41] thus LIFR blockade alone may not be sufficient to inhibit these cytokines. These data suggest, in combination with the toxicity study performed with mLIFR-Fc, that treatment with eLIFR-Fc would have minimal off-target effects and limited on-target toxicity.

## eLIFR-Fc affects growth progression of human PDAC cells in a mouse tumor model.

To establish a model to test the therapeutic efficacy of eLIFR-Fc in vivo, we evaluated conditioned media collected from cultured KP4 human PDAC cells by ELISA and found an appreciable level of LIF was being produced (Supplementary Fig. 6e). In addition, KP4 cells generated a strong pSTAT3 signal and were susceptible to subsequent LIF inhibition when supplemented with LIF (Fig. 4c, d). While some human PDAC cells secrete LIF in an autocrine loop, much of the LIF in the tumor microenvironment is derived from the surrounding activated stellate cells[3,4]. These pancreatic myofibroblast cells and any secreted LIF they produce are therefore largely missing from flank xenograft models of human cancer, where any mouse-derived LIF that is produced will not bind and activate human LIFR[43].

In nude mice with KP4 flank xenograft tumors, we compared intraperitoneal injection of PBS vehicle to treatment with eLIFR-Fc (10 mg/kg, 2 times per week). KP4 tumors treated with eLIFR-Fc therapy showed significantly slowed tumor growth (Fig. 4f, Supplementary Fig. 6f–h), with excised tumors at the end of the study confirmed to be significantly smaller than those from the saline treated control group (Supplementary Fig. 6f, g). These data indicate that eLIFR-Fc is a promising treatment approach for further development.

## Discussion

LIF promotes aggressive tumor growth, and PDAC patients with high *LIF* mRNA levels have much worse overall survival than those with low *LIF* mRNA levels[3,44]. To effectively target LIF in human PDAC progression, we engineered a soluble version of the hLIFR extracellular domain as a ligand trap (Fig. 1a). We identified eight mutations (I217V, L218P, H240R, I257V, I260V, V262A, T273I, and N277D) (Fig. 2e) that collectively increase the affinity of soluble eLIFR-Fc over hLIFR-Fc by ~50-fold (Fig. 3d). This construct is among the highest affinity LIF-binding agents known to date. To infer how these mutations improved affinity, we modeled the structure of eLIFR binding to hLIF and mapped the mutations to examine their locations (Materials and Methods). Mutations were clustered in three loops of the Ig-like domain (Supplementary Fig. 7a). Several mutations appear in the interface (e.g. H240R, T273I, and N277D), while others such as I217V, I257V, and I260V suggest possible subtle changes in the

structure. These changes do not result in improved thermal stability (Supplementary Fig. 3e), indicating that the improved properties demonstrated by eLIFR-Fc are likely not the result of improved stability over hLIFR-Fc. Both LIFR-Fc proteins displayed minimal aggregation (Supplementary Fig. 3d,e), even at high temperatures, a favorable quality for therapeutic development, and in stark contrast to a control IgG, which formed large aggregates at higher temperatures (Supplementary Fig. 3e).

LIFR possesses interesting unidirectional cross-species interactions: mLIFR binds both mLIF and hLIF, but hLIFR only binds hLIF. Intriguingly, mLIFR binds hLIF with very high affinity (~1 pM) (Supplementary Fig. 7b), inhibits hLIF signaling as a soluble monomer, and can also be readily expressed as a soluble Fc-fusion[30]. While mLIFR-Fc is a potent and effective hLIF inhibitor, it has only ~70% sequence homology to hLIFR. Strikingly, by introducing eight mutations into hLIFR to create eLIFR, we boosted the affinity of the human receptor such that it also binds hLIF with low pM affinity (Fig. 3d, Supplementary Fig. 3c). Notably, three of the mutations in eLIFR are residues found in the mLIFR sequence (I217V, T273I, and N277D). The remaining five mutations are unique and likely could have only been discovered through combinatorial protein engineering efforts performed here. Further improvements in affinity and species cross-reactivity may thus be possible through insertion of other regions of mLIFR that contact LIF (such as the CBM II) into the eLIFR scaffold; however, additional changes may result in increased immunogenicity.

Blockade of ligand binding to extracellular receptors, traditionally through neutralizing antibodies, has been widely used as a therapeutic strategy against cancer[45,46]. Ligand traps, also known as receptor decoys, have emerged as effective alternatives to antibodies. The native interactions between ligands and their cognate cell surface receptors are often of very high affinity[47,48]. LIF is no exception, as the binding of LIF with the LIFR/gp130 receptor complex is reported to be in the 50–100 pM range[47,48]. Monoclonal antibodies typically bind their targets with affinities in the low nM range[45], although the D25 mAb binds to mouse and human LIF with apparent affinities in the mid-pM range (Supplementary Fig. 1c and 3c). Given the high-affinity LIF–LIFR–gp130 complex, the engineered eLIFR-Fc ligand trap more effectively inhibits LIF binding and activity as compared to hLIFR-Fc. These results are aligned with our previous work showing that an Axl receptor decoy engineered to bind its ligand, Gas6, with femtomolar affinity was more effective than the WT Axl receptor decoy in blocking the low picomolar interaction between Gas6/Axl that drives metastatic spread of aggressive cancers[27,28]. The benefits of treatment with a ligand trap versus an antibody remain understudied; the compounds developed here provide a robust set of tools for exploring this question in a future study.

While the bivalent design of eLIFR-Fc is attractive due to ease of production and benefits of avidity as indicated in this study, we found that monovalent 'one-arm' eLIFR-Fc is also a potent LIF inhibitor. Monovalent eLIFR-Fc is nearly as efficacious as eLIFR-Fc in competing LIF away from WT LIFR (Fig. 3e) and gp130 (Fig. 3f) and inhibiting LIF signaling in HeLa STAT3 luciferase reporter cells (Supplementary Fig. 4d, e). The one-arm version of eLIFR-Fc is much smaller than the bivalent eLIFR-Fc (~107 kDa versus ~173 kDa; Supplementary Fig. 3b), and could be a viable alternative if needed for navigating the dense stroma of the tumor microenvironment, though as LIF is a secreted factor, deep tumor penetration may not be a prerequisite for therapeutic efficacy.

eLIFR-Fc potently competes LIF away from both WT LIFR (Fig. 3e) and, somewhat surprisingly, gp130 (Fig. 3f). This unexpected feature might be due to a second, but minor LIF-binding site on LIFR, which may compete with gp130 for

binding[49,50]. It is also possible that the soluble ligand trap induces a conformational change in LIF that no longer permits gp130 engagement. Although the D25 antibody binds LIF with high affinity on the LIFR binding face, it does not compete with gp130 for binding. As the one-arm eLIFR-Fc fusion also potently competes LIF away from LIFR and gp130 (Fig. 3f), this property may be innate to LIFR and could be an evolved mechanism, as soluble LIFR is naturally secreted as an inhibitor[51,52]. Such dual-specific competition of both LIFR and gp130 could potentially contribute to eLIFR-Fc inhibition, for example, if LIF binding to cell surface gp130 in the absence of LIFR binding still increases the local concentration of soluble factor.

The present investigations underscore the importance of LIF as an oncogenic factor. Based on our results and the work of others, it is likely that LIF reprises its stem-factor role in cancer, promoting the survival of cancer stem cells to increase sphere formation and tumor growth[3,4,6]. Further, as a cytokine, LIF is capable of supporting signaling in multiple types of immune cells and has recently been shown to be immunosuppressive in glioblastoma, highlighting a potential role for LIF inhibition as part of an immunotherapy regimen[53]. LIF is also a well-documented cachexia factor in multiple cancers, particularly PDAC, dramatically affecting quality of life and occasional morbidity[54–56]. In future studies, it would be interesting to explore the role of eLIFR-Fc and the D25 mAb on immunosuppression and cachexia in animal models as an additive benefit to tumor control. By targeting LIF it may be possible to block or minimize features that render PDAC so devastating, highlighting the urgency and importance of developing eLIFR-Fc and other LIF inhibitors for additional pre-clinical and clinical studies.

## Methods

**KPC mouse treatment and pancreas immunohistochemistry.** $Kras^{G12D/+}$; $Trp53^{f/f}$; $Rosa26^{Luc}$; $Pdx1$-$Cre$ (KP$^{f/f}$CL) mice were generated and maintained as described previously[3]. Recombinant mLIFR (CBM I–CBM II: Met1 to Thr529) was produced in mammalian cells as described below. The anti-LIF antibody D25 was obtained from the Hunter Lab, produced from a hybridoma line (HB-11074, ATCC) and collected from mouse ascites, as described previously[3]. For the pSTAT3 inhibition study, 5-week-old KP$^{f/f}$CL male and female mice were intraperitoneally administered three doses of 20 mg/kg mLIFR-Fc (days 1, 3, and 5). Control mice were given an isotype control IgG or the D25 mAb at 25 mg/kg, three times (days 1, 3, and 5) intraperitoneally. On day 7, mice were sacrificed and the pancreases were collected, fixed in 10% neutral-buffered formalin overnight at 4 °C, and paraffin embedded according to standard protocol.

For immunohistochemistry, 5-μm tissue sections were deparaffinized in xylene and rehydrated in graded ethanols. Antigen retrieval was performed for 15 min in 95–100 °C 10 mM sodium citrate buffer [pH 6.0] or 1 mM EDTA [pH 8.0] (for pSTAT3 antibody). Then, endogenous peroxidase activity was quenched with 3% $H_2O_2$ for 10 min. Sections were blocked in TBS containing 0.1% Triton X100 (Sigma) and 5% goat or donkey serum (Vector Laboratories). Incubations with primary antibodies were performed overnight at 4 °C in a humidified chamber, followed with appropriate SignalStain Boost IHC Detection Reagent (Cell Signaling Technology) for 30 min, or with biotinylated secondary antibodies for 45 min and then ABC Elite for 30 min (Vector Laboratories), all at room temperature. ImmPACT DAB Kit (Vector Laboratories) was used to develop signals in accordance with the manufacturer's instructions. Sections were counterstained with hematoxylin (Sigma). Rabbit anti-Cytokeratin 19 (Epitomics AC-0073; 1:200) was used for PDAC lesion detection. Rabbit anti-phospho-STAT3 (Tyr705) (9145, CST; 1:100) was used for pSTAT3 detection.

**Mouse toxicity studies.** Friend Virus B NIH Jackson (FVB/NJ) (001800, The Jackson Laboratory) and Black/6 (C57B/6J) (000664, The Jackson Laboratory) 5–6-week-old female mice were used to study protein toxicity. Mice were injected with 20 mg/kg of mLIFR-Fc or PBS intraperitoneally, three times per week, for 4 weeks. Upon each administration, mice were weighed. At the end of the study, a full necropsy was performed by the Stanford Veterinary Service Center. Mice ($n = 16$ total; $n = 8$ FVB and $n = 8$ C57BL/6) were euthanized via CO$_2$ asphyxiation and cardiac exsanguination. Terminal cardiac blood was collected for complete blood counts and serum biochemistry. All mice were weighed, and complete gross examination of the external and internal organs was performed. All tissues were collected and immersion-fixed in 10% neutral-buffered formalin for 72 h. Formalin-fixed tissues from FVB mice ($n = 8$) were processed routinely, embedded in paraffin, sectioned at 5 μm, and stained with hematoxylin and eosin. The

following tissues were examined histologically: heart, liver, spleen, kidneys, adrenal glands, submandibular salivary glands, submandibular lymph nodes, pancreas, thymus, tongue, oropharynx, esophagus, trachea, lungs, haired skin, brown fat, brain, reproductive tract (uterus, ovaries, cervix, vagina), and gastrointestinal tract (stomach, duodenum, jejunum, ileum, cecum, proximal colon, distal colon, rectum). All slides were blindly evaluated by a board-certified veterinary pathologist.

**Yeast-surface display binding assays.** DNA sequences encoding proteins were cloned into the pCTCON2 yeast-surface display vector (41843, Addgene) using the NheI and BamHI sites. Proteins displayed on yeast include: hLIF (Ser23 to Phe203), hLIFR (CBM I–CBM II: Gln45 to Ser534; CBM I–Ig-like: Gln45 to Gln337), and mLIF (Ser24 to Phe203). EBY100 yeast were transformed with pCTCON2 plasmids and selected on SD-CAA-Agar plates. Yeast (~100,000 per sample) were grown and induced, and binding set up as described previously[57] over a range of soluble ligand or receptor concentrations in PBS containing 1 mg ml$^{-1}$ bovine serum albumin (BSA; BPBS), taking into account ligand depletion and equilibrium time. After incubation with binding partner, yeast cells were washed once with BPBS, then incubated with a 1:5000 dilution of chicken anti-c-myc antibody (A21281, Invitrogen). For His$_6$-tagged proteins, a 1:500 dilution of rabbit anti-6-His-FITC antibody (NC0585792, Fisher Scientific [Bethyl Labs]) was added. All primary antibodies were incubated for 30 min at 4 °C in the dark. After primary addition, samples were washed once with BPBS, and secondary antibodies were added. Expression was detected with a 1:500 dilution of goat anti-chicken Alexa Fluor 488 (A11039, Fisher Scientific), a 1:500 dilution of goat anti-chicken phycoerythrin (PE; sc-3730, Santa Cruz Biotechnology), or a 1:500 dilution of goat anti-chicken Alexa Fluor 647 (NC0928213, Fisher Scientific [Abcam]). Binding of His$_6$-tagged proteins was detected with a 1:500 dilution of goat anti-rabbit Alexa Fluor 488 (A11034, Fisher Scientific). Binding of proteins with the mIgG2a Fc tag was detected with a 1:500 dilution of goat anti-mouse Alexa Fluor 488 (A11029, Fisher Scientific) or a 1:500 dilution of goat anti-mouse Alexa Fluor 647 (A21463, Fisher Scientific). To detect binding of proteins with the hIgG1 Fc tag, a 1:500 dilution of goat anti-human Alexa Fluor 647 (A21445, Fisher Scientific) was used. Secondary antibodies were incubated for 15 min at 4 °C in the dark. After secondary incubation, samples were washed once with BPBS, pelleted, and left pelleted on ice until analysis. Samples were analyzed by resuspending them in 50 μL of BPBS and running flow cytometry using a BD Accuri C6 (BD Biosciences).

Samples were gated for bulk yeast cells (forward scatter (FSC) vs. side scatter (SSC)) and then for single cells (FSC-Height vs. FSC-Area). Expressing yeast were determined and gated via C-terminal c-myc tag detection. The geometric mean of the binding fluorescence signal was quantified from the expressing population and used as a raw binding value. When comparing binding signals, the average fluorescence expression signal was quantified for different protein variants and used to normalize binding signal. To determine "fraction bound," binding signals were divided by the signal derived from the highest concentration of binding partner used. To calculate $K_d$ values, data were analyzed in GraphPad Prism (v8.0.2) using non-linear regression curve-fit.

**Generation and screening of a LIFR library created via error-prone PCR.** LIFR was expressed in *Saccharomyces cerevisiae* (strain: EBY100; ATCC MYA-4941) as a genetic fusion to the agglutinin mating protein Aga2p. cDNA encoding the human LIFR CBM I–Ig-like extracellular domains (Gln45 to Gln337) was cloned into the pCTCON2 yeast display plasmid (41843, Addgene) through the NheI and BamHI restriction sites (Forward primer: 5′-ATATAGGATCCTTGAGGAGTAT CTGGTGGAT-3′; Reverse primer: 5′-ATATAGCTAGCCAGAAAAAGGGGG CTCCTCA-3′). An error-prone library was created using the LIFR Ig-like domain (Gln251 to Gln337) as a template and mutations were introduced with Taq polymerase (50-811-694, Fisher Scientific), 50 mM MgCl$_2$, and nucleotide analogs (Forward primer: 5′-TGTGAAGAACATTTCTTGGATACCTGATTCT-3′; Reverse primer: 5′-GATCTCGAGCTATTACAAGTCCTCTTCAGAAATAAGCT TTTGTTCGGATCCA-3′). Separate PCRs were performed using various concentrations (0.5, 1, 2.5, 5, 10, and 20 μM) of the dNTP analogs 2′-dPTP (N-2037, TriLink Biotechnologies) and 8-oxo-2′-dGTP (N-2034, TriLink Biotechnologies). Products from these reactions were purified via gel electrophoresis, pooled, and amplified with standard PCR using Phusion polymerase (M0530S, New England BioLabs). Purified mutant cDNA and linearized plasmid were electroporated into EBY100 yeast, where they were assembled in vivo through homologous recombination. We estimated $2.7 \times 10^7$ variants for the first-generation library and $1.6 \times 10^8$ variants for the second-generation library, determined by dilution plating and colony counting. The second-generation library was created from DNA isolated from library 1, sort round 4 which was subjected to error-prone PCR (as described above) and DNA shuffling using DNAse I as previously described[58].

Yeast were grown and induced for LIFR protein expression as previously described[57]. Yeast displaying high-affinity LIFR variants were isolated via FACS using a BD Aria II flow cytometer (Stanford FACS Core Facility) and analyzed with a BD Accuri C6 flow cytometer (BD Biosciences). Data were analyzed using FlowJo software (v10.6.1, Tree Star Inc.). Screens were carried out using a mixture of equilibrium binding and kinetic off-rate conditions where yeast were incubated at room temperature in BPBS with the following concentrations of LIF-His. Library 1: Sort 1, 1 nM LIF for 10 h; Sort 2, 100 pM LIF for 12 h; Sort 3, kinetic off-rate, incubate LIF-bound yeast in 2 mL BPBS with 10 nM untagged LIF competitor for

16 h; Sort 4, kinetic off-rate, incubate LIF-bound yeast in 2 mL BPBS with 10 nM untagged LIF competitor for 36 h; Sort 5, kinetic off-rate, incubate LIF-bound yeast in 50 mL BPBS with 3 nM untagged LIF competitor for 36 h; Sort 6, kinetic off-rate, incubate LIF-bound yeast in 50 mL BPBS with 5 nM untagged LIF competitor for 36 h. Library 2: Sort 1, 200 pM LIF for 20 h; Sort 2, 100 pM LIF for 17 h; Sort 3, 10 pM LIF for 31 h; Sort 4, kinetic off-rate, incubate LIF-bound yeast in 100 µL BPBS with 675 nM untagged LIF competitor for 40 h; Sort 5, kinetic off-rate, incubate LIF-bound yeast in 100 µL BPBS with 2 µM untagged LIF competitor for 16 h. After incubation with LIF-His, yeast were pelleted, washed, and labeled with fluorescent antibodies as described above prior to sorting.

Sorted yeast clones were propagated, induced for LIFR expression, and subjected to iterative rounds of FACS as described above. After each round of screening, plasmid DNA was recovered using a Zymoprep yeast plasmid miniprep I kit (D2001, Zymo Research Corp), transformed into DH10B electrocompetent cells (18297010, Thermo Fisher), and isolated using a GeneJET plasmid miniprep kit (K0503, Thermo Fisher). Sequencing was performed by Molecular Cloning Laboratories (South San Francisco, CA).

**Generation of LIFR variants**. LIFR variants were created from existing isolated plasmid DNA using site-directed mutagenesis. Briefly, forward and reverse primers were designed to incorporate the desired mutation, surrounded by flanking sequences of 20 base pairs with homology to the parent LIFR gene. PCR was performed on each LIFR plasmid over two successive rounds: first with the forward and reverse primers alone, and then in a pooled reaction. PCR product was treated with DpnI (R0176S, New England Biolabs) for 2 h at 37 °C, followed by enzyme denaturation at 80 °C for 20 min. Five microliters of the digested product was transformed into DH10B cells and the plasmid DNA was isolated using a GeneJET plasmid miniprep kit (K0503, Thermo Fisher). Sequencing was performed by Molecular Cloning Laboratories.

**Calculating binding scores for LIFR variants**. LIFR variants were displayed on yeast as described above. Variants were incubated with two low concentrations of LIF (100 and 10 pM) and allowed to come to equilibrium (~48 h). Binding signal was determined in each case, as described above. Raw fluorescent binding signal of each variant was summed at each concentration of LIF, and normalized to the expression of that variant. This value was divided by the value generated from WT hLIFR to calculate a "binding score" for each mutant variant.

**Modeling of the LIF-LIFR interaction using Rosetta**. The LIF–LIFR binding interaction was modeled using RosettaRemodel to introduce mutations, and RosettaDock to locally dock and propose the binding interface of hLIF–eLIFR (Supplementary Fig. 7c). Solved crystal structures of hLIFR CBM I–CBM II (PDB: 3E0G) and the hLIF–mLIFR complex (PDB: 2Q7N) were used as templates. First, PDB structures were prepared for Rosetta by relaxing with all-heavy atom constraints. The eight "VPRVVAID" mutations were then modeled into the hLIFR structure using RosettaRemodel, bypassing fragment insertion and running three design-relaxation cycles on-target amino acids. The two flanking residues of each target amino acid, as well as neighboring residues (6 Å), were allowed to be repacked without altering their amino acid identity. The lowest energy structure from 1000 trajectories was chosen to move forward with for docking analysis and was deemed the eLIFR apo-structure. As a starting position for local docking, the site of the hLIF–eLIFR interaction was determined from the hLIF–mLIFR complex structure. 10,000 trajectories of local docking were run between hLIF and the eLIFR structure, and the lowest interface energy structure was identified as the most likely interaction (Supplementary Fig. 7a). Of these 10,000 trajectories, there was a strong convergence via RMSD to the lowest energy model (Supplementary Fig. 7d).

**Recombinant protein expression**. cDNA corresponding to hLIFR (CBM I–CBM II: Met1 to Ser534; full length: Met1 to Ser833), hLIF (Met1 to Phe203), and mLIFR (CBM I–CBM II: Met1 to Thr529) were cloned into the pAdd2 plasmid[28,29] to contain a hexahistidine tag, a mIgG2a mFc domain, or an hIgG1 hFc domain. For the one-arm monovalent Fc fusions, the 'knobs-into-holes' hIgG1 mutations (CH3$_A$: T366W; CH3$_B$: T366S, Y407V, L368A) were used. LIFR was cloned into the pAdd2 vector containing the CH3$_A$ sequence, while the CH3$_B$ sequence did not include a fusion gene. Dinutuximab was cloned into the pAdd2 plasmid and purified as a control IgG. pAdd2 genes were amplified in DH10B bacterial cells and DNA was collected using a PureLink HiPure plasmid filter maxiprep kit (K210016, Fisher Scientific).

For expression, purified plasmids were transfected into Expi293F cells (A14527, Fisher Scientific) using the Expifectamine 293 Transfection Kit (A14525, Fisher Scientific). Briefly, for a 120-mL transfection, 120 µg DNA was incubated in 5.9 mL of Opti-MEM I reduced serum media (31-985-088, Fisher Scientific), while 318 µL of Expifectamine was incubated in another 5.9 mL of Opti-MEM for 5 min at room temperature. These two samples were mixed gently and incubated for an additional 20 min. All 11.8 mL were then added dropwise to ~100 mL of Expi293F cells grown in Expi293 Media (A1435102, Fisher Scientific; density $2.9 \times 10^6$ cells/mL). Cells were grown as recommended by the manufacturer. After 18 h, 590 µL of Enhancer 1 and 5.9 mL of Enhancer 2 (both from the Expifectamine 293 Transfection Kit) were added. Cells were left to produce protein for an additional 5 days at 37 °C.

When expression was complete, medium was collected via centrifugation and purified using Protein-A Sepharose beads (101142, Fisher Scientific) for Fc-tagged proteins or Ni-NTA agarose beads (30230, Qiagen) for His$_6$-tagged proteins. Proteins were concentrated and buffer-exchanged into PBS using Amicon Ultra-15 centrifugal filter units (UFC905024, Millipore-Sigma; 50 kDa cut-off), and subsequently stored at 4 °C. Concentration was calculated using $A_{280}$ on a NanoDrop 2000 (Thermo Fisher), with the following extinction coefficients: LIFR hIgG1 Fc variants: 297,000 M$^{-1}$ cm$^{-1}$; LIFR mIgG2a variants: 219,040 M$^{-1}$ cm$^{-1}$; LIFR one-arm Fc variants: 174,190 M$^{-1}$ cm$^{-1}$; LIFR Full Length hIgG1 Fc variants: 416,345 M$^{-1}$ cm$^{-1}$; mLIFR mIgG2a Fc: 311,995 M$^{-1}$ cm$^{-1}$; hLIF: 10,805 M$^{-1}$ cm$^{-1}$.

Recombinant hLIF-His (14890-H08H-20, Sino Biological Inc.), hLIF (untagged) (14890-HNAH-50, Sino Biological Inc.), mLIF-His (ABIN2215872, Antibodies-Online), hOSM-His (10425-H08H-20, Sino Biological Inc.), and hCTF-1-Fc (16013-H01H-20, Sino Biological Inc.) were purchased for use. All purchased proteins were produced in mammalian cells by the manufacturer and contained the His$_6$ tag or Fc fusion on their C-termini. hCLCF-1-FLAG and hCNTFR-His were produced as described previously[29]. The D25 and antibody was received as a gift from the Hunter lab, having been produced from a hybridoma line (HB-11074, ATCC) and collected from the ascites of mice.

**Size exclusion chromatography**. eLIFR-Fc and hLIFR-Fc were analyzed by size exclusion chromatography on an Agilent 1260 Infinity II Analytical-Scale LC Purification System using a Superdex 200 3.2/300 column (GE28-9909-46, Millipore-Sigma).

**Melting and aggregation temperature measurements using the Uncle**. An Uncle instrument (Unchained Labs) was used to measure full spectrum fluorescence and SLS at 266 nm (SLS$_{266 nm}$) over temperatures ranging from 15 to 95 °C, measuring at a ramp rate of 0.5 °C/min. DLS measurements were also taken at the initial (15 °C) and final (95 °C) temperatures. Fluorescence, SLS, and DLS data were analyzed using Uncle Analysis software to generate $T_m$, $T_{agg}$, and average particle size measurements (Z-average), respectively.

**Binding measurements using the Kinetic Exclusion Assay (KinExA)**. A KinExA 3200 instrument (Sapidyne Instruments Inc.) was used to measure equilibrium binding of LIFR-Fc and LIF. Polymethyl methacrylate beads (98 micron, PMMA beads; 440107, Sapidyne Instruments) were coated with eLIFR-Fc or hLIFR-Fc and used to detect free LIF-His. Adsorption coating of 200 mg of beads was performed using 40 µg of eLIFR-Fc or hLIFR-Fc at room temperature for 2 h. The protein solution was removed, and beads were blocked using PBS with 10 mg/mL BSA at 4 °C overnight. Beads were stored in blocking buffer at 4 °C and used within 3 days.

Soluble hLIF-His was incubated at a constant concentration with serially diluted soluble eLIFR-Fc or hLIFR-Fc in KinExA Running Buffer (BPBS with 0.02% sodium azide). For eLIFR-Fc binding, LIF was held at a constant concentration of 100 pM, while for hLIFR-Fc binding, LIF was held at a constant concentration of 250 pM. eLIFR-Fc was diluted to a concentration of 9 nM in 3 mL, then serially diluted 1:3 over 11 more tubes (final concentration 50 fM). hLIFR-Fc was diluted to a concentration of 27 nM in 2 mL, then serially diluted 1:3 over 11 more tubes (final concentration 150 fM). Protein-binding reactions were incubated for 18 h before being analyzed on the KinExA 3200 as an equilibrium assay. Binding of free LIF-His to eLIFR- or hLIFR-coated beads was detected using an anti-6xHIS Dylight 649 antibody (200-343-382, Rockland Immunochemicals Inc.). Each sample was measured twice, and the data were globally analyzed using n-curve analysis with KinExA Pro 3.6.2 software (Sapidyne Instruments Inc.) to obtain the apparent $K_d$ values.

**Yeast display competition-binding measurements**. hLIF was displayed on yeast as described above. hLIFR-Fc or eLIFR-Fc was incubated with yeast displaying LIF at saturating concentrations (10 nM) for 1 h. Yeast were washed once with BPBS, then resuspended in buffer containing no hLIF, 10 nM hLIF, or 100 nM hLIF. Samples were incubated at room temperature overnight, and Fc-fusion binding to yeast-displayed LIF was quantified as described above. Each variant was normalized to the wash condition without soluble LIF added.

WT LIFR or gp130 was displayed on yeast as described above. hLIF was added at saturating concentrations (10 nM) and incubated for 1 h at room temperature. Without washing samples, 100 nM of hLIFR-Fc, eLIFR-Fc, one-arm eLIFR-Fc, or the D25 mAb were added. Samples were incubated for 3 days at room temperature. Binding of LIF to yeast-surface displayed LIFR or gp130 was quantified as described above. Binding signal was normalized to a no-inhibitor control condition for each displayed receptor.

**Cell lines**. PANC1, PANC1 LIFR knockdown (KD), KP4, KP4 LIFR knockdown (KD), and mPDAC cells were provided by the Hunter group (Salk Institute) and cultured as previously described[3]. STAT3 Luciferase Reporter HeLa Stable cells (SI-0003-NP) were purchased from Signosis, and cultured in accordance with the supplier's instructions.

**HeLa STAT3 firefly luciferase reporter cell assays**. For all assays, cells were seeded into 96-well plates at a density of $1.5 \times 10^4$ cells/well. Prior to cytokine addition, cells were switched to serum-free Dulbecco's Modified Eagle Medium (DMEM) for 2–6 h. Assays were carried out in DMEM supplemented with 0.1% fetal bovine serum (FBS; HeLa assay medium). After incubation with cytokine and/ or ligand traps, luciferase activity was quantified using the Luciferase Assay System (E4030, Promega). Briefly, cells were washed once with PBS and lysed with 20 μL of 1× Reporter Lysis Buffer (E397A, Promega). Luciferase assay substrate (E151A, Promega) was resuspended in luciferase assay buffer (E152A, Promega), and 100 μL were added to each well containing cell lysate. Luminescence was measured for 2 s using a BioTek Synergy H4 plate reader. All experiments were completed in triplicate.

The dynamic range of LIF signaling in HeLa STAT3 cells over 4 h was determined by serial dilution (starting at 30 nM LIF, diluted 10 times by 1:3 to a concentration of 0.5 pM). Inhibition was determined by incubating eLIFR-Fc, one-arm eLIFR-Fc, or hLIFR-Fc at varying concentrations (0.5–20 nM) in the presence of 0.5 nM LIF for 1 h at room temperature. Proteins were added to serum-starved HeLa cells for 4 h at 37 °C. For $IC_{50}$ analysis, eLIFR-Fc or hLIFR-Fc were serially diluted (eLIFR: 30 nM to 1.5 pM; hLIFR: 300 nM to 15 pM) in HeLa assay medium with a constant concentration of 20 pM LIF, reflective of the possible concentration of LIF in the pancreatic cancer microenvironment[3]. Samples were incubated overnight at room temperature, then added to HeLa cells for 5 h at 37 °C before measuring luciferase signal. For delayed addition, LIF was added to HeLa cells in HeLa assay buffer at a concentration of 0.5 nM and left for 1 h at 37 °C. eLIFR-Fc or hLIFR-Fc was added at a concentration of 5 or 25 nM and incubated for 20 h at 37 °C before measuring luciferase activity. Assays with other cytokines (0.5 nM OSM, 3 nM CLCF-1, 3 nM CTF-1) were carried out in the same manner as assays with LIF.

**PDAC cell signaling assays**. KP4, KP4 LIFR KD, PANC1, PANC1 LIFR KD, and mPDAC cells were seeded into 12-well plates at a density of $0.1 \times 10^6$ cells/well. Assays were performed when cells were 75% confluent. Prior to cytokine addition, cells were starved in serum-free medium for 2 h at 37 °C. For the LIF time-course, LIF was diluted to 0.5 nM in serum-free medium supplemented with 0.1% BSA, and then added to cells for varying amounts of time (from 1 to 60 min), compared to addition of medium alone. For inhibition assays, eLIFR-Fc, hLIFR-Fc, mLIFR-Fc, or the D25 mAb were added at varying concentrations from 0.05 to 500 nM to serum-free medium supplemented with 0.1% BSA and 0.5 nM LIF and incubated for 1 h at room temperature. Samples were added to serum-starved cells for 20 min, along with samples containing no LIF, LIF with no inhibitor, or medium supplemented with 10% FBS ("Serum"; 26-140-079, Fisher Scientific).

After incubation, cells were washed twice with ice-cold PBS. After washing, 100 μL of RIPA Lysis and Extraction Buffer (89900, Fisher Scientific) supplemented with phosphatase inhibitor (P5726, Sigma-Aldrich) and protease inhibitor (P8340, Sigma-Aldrich) were added to each well, and cells were manually lysed by scraping with the back of a p1000 pipette tip. Lysed cells were incubated in RIPA buffer on ice for 10 min. Lysates were then collected, transferred to Eppendorf tubes, and sonicated in an ultrasonic water bath (Branson 1800) for 10 s. Samples were centrifuged at $16,100 \times g$ at 4 °C for 10 min to clear precipitate. Supernatant was collected and protein concentration was quantified using a Pierce BCA Protein Assay Kit (23225, Fisher Scientific).

Twenty micrograms of protein from each lysate was added to a solution containing 1× NuPAGE LDS Sample Buffer (NP0007, Fisher Scientific) and 1× NuPAGE Sample Reducing Agent (NP0009, Fisher Scientific). Samples were boiled for 10 min before they were loaded onto a 15-well, 4–20% Expressplus PAGE gel (M42015, GenScript). Gels were run for 45 min at 160 V, before samples were transferred to a nitrocellulose membrane (IB301031, Thermo Fisher) using an iBlot dry blotting system (IB1001, Thermo Fisher). Blots were blocked in Tris-buffered saline, pH 7.4, with Tween-20 (TBST) with 5% BSA for 1 h at room temperature. STAT3 was detected using Stat3 (D3Z2G) Rabbit mAb #12640 (NC0969631, Fisher Scientific) at 1:5000 in TBST plus 5% BSA. pSTAT3 was detected with phospho-stat3 (pY705) Rabbit mAb (9145S, Fisher Scientific) at 1:4000 in TBST plus 5% BSA. As a protein loading control, tubulin was detected using anti-B-tubulin (TU27/Tubulin) Mouse antibody (50-103-0148, Fisher Scientific). STAT3 and pSTAT3 were stained on separate blots: sample from the same stock was added to each gel and the gels were run and transferred simultaneously. Primary antibody was incubated with blots for 2 h at room temperature. Blots were then washed three times for 5 min in TBST before secondary antibody was applied. Secondary antibodies used were Peroxidase-AffiniPure Donkey Anti-Mouse IgG (tubulin antibody; 715-035-150, Jackson ImmunoResearch) and Peroxidase-AffiniPure Donkey Anti-Rabbit IgG (stat3 antibodies; 711-035-152, Jackson ImmunoResearch) at 1:5000 dilutions in TBST plus 5% BSA. Secondary antibody was incubated with the blot for 1 h at room temperature. Blots were washed three more times with TBST for 5 min each, exposed to SuperSignal West Femto Maximum Sensitivity Substrate (34095, Fisher Scientific), and chemiluminescence was detected using the ChemiDoc XRS System (Bio-Rad). Images were quantified using ImageJ (1.49 v). Assays with OSM (0.5 nM) were carried out in the same manner as those with LIF.

**Assays of sphere formation by PDAC cells**. Sphere-formation assays were carried out in DMEM/F12 medium (11330-032, Fisher Scientific) supplemented with 20 ng/mL FGF-Basic (PHG0026, Fisher Scientific), 1× B27 supplement (12587010, Fisher Scientific), and 0.75% methylcellulose (M-352, Fisher Scientific). Cells were seeded into 96-well ultra-low attachment plates (3474, Corning) at a density of 100 cells/well. LIF was added at 250 ng/mL (12 nM). eLIFR-Fc or hLIFR-Fc was added at 25, 150, or 500 nM. Cells were allowed to grow as spheres for 1–2 weeks before each whole well (4× zoom) was imaged on an Incucyte (Satorius). In ImageJ (1.49 v), spheres were identified by shape and size, circled, and the total area of spheroids in each well was calculated. Experiments were repeated over six wells in each condition.

**Determination of LIF levels in conditioned media using ultra-sensitive ELISA**. KP4 cells were seeded into a six-well plate at $0.3 \times 10^6$ cells/well. Cells were grown to ~75% confluency, then switched into RPMI + 1% FBS media and left for 24 h. Media was collected, spun down at 3000 rpm for 5 min to remove debris, and frozen at −80 °C. LIF levels were determined by ultra-sensitive ELISA, as described previously[3].

**Xenograft study with KP4 cell line**. KP4 cells ($5 \times 10^6$) were engrafted into the right flank of 6-7 weeks old nude, female mice (NU/J, 002919, The Jackson Laboratory; $n = 25$ mice). Cells were resuspended in PBS before being mixed 1:1 with Matrigel (CB-40234A, Fisher Scientific) and injected using 100 μL of cell/ Matrigel mixture per mouse. Tumors were allowed to grow for 20 days before mice were separated into treatment groups ($n = 7$ mice per arm). Only mice with firmly engrafted tumors >130 mm[3] were separated into treatment groups. Tumor sizes were evenly distributed between groups. Mice were treated with PBS or eLIFR-Fc (10 mg/kg), given 2×/week for 2.5 weeks. Mice were weighed prior to compound administration and throughout the study. Tumors were measured with digital calipers 2×/week at the same time as dosing. Study was concluded when tumor measurements surpassed euthanasia criteria. At the conclusion of the study, tumors were excised, weighed, and measured using digital calipers.

**Statistics and reproducibility**. The specific statistical tests utilized are indicated in the figure legends. Statistical analyses were performed using Prism (v8.0.2, GraphPad Software). For comparisons between two groups, statistical significance was assayed by two-tailed unpaired Student's $t$ test. For comparison within in vivo studies, a two-way analysis of variance (ANOVA) combined with Sidak's multiple comparison test for post hoc analysis was performed. Sample sizes were determined on the basis of the variability of pancreatic tumor models used. Tumor-bearing animals were assigned to the treatment groups to ensure an equal distribution of tumor sizes between groups. Data are represented as mean ± standard deviation (in vitro studies) or mean ± standard error of the mean (in vivo studies). For all statistical analyses, P values are indicated in the figure legend.

**Ethics statement**. Mice were maintained and animal experiments performed in accordance with policies approved by the Stanford University Administrative Panel on Laboratory Animal Care (Protocol no. 33187) and the Salk Institute Animal Care and Use Committee.

**Reporting summary**. Further information on research design is available in the Nature Research Reporting Summary linked to this article.

## Data availability

The data that support the findings of this study are available from the corresponding author upon reasonable request. Source data for the main figures is included in the supplementary data.

## Code availability

The custom scripts for identification of mutations in the sorted library from sequencing data and the modeling of mutations in LIFR and docking of the LIF–LIFR interaction using Rosetta will be freely available upon reasonable request.

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

## Acknowledgements
We thank all members of the Cochran lab for helpful suggestions, particularly A.C. Mitchell, G.M. Cherf, and J.W. Kim for discussions, practical advice, and protocols. We

thank P. Huang for advice and feedback during the modeling of the eLIFR-LIF inter-action. We thank J. Silberstein for reagents and valuable feedback. We also thank S. Yamada, P. Jackson, J. Sage, and T. Vora for helpful feedback during manuscript preparation. Sorting was performed on instruments in the Stanford Shared FACS facility, including an instrument obtained using an NIH S10 Shared Instrument Grant (S10RR025518-01), and we thank the Stanford Shared FACS Facility for advice and assistance during the yeast library sorting process. We thank the Stanford Veterinary Service Center for advice in conducting mouse experiments and in performing toxicity analysis studies. S.A.H. was supported by a National Science Foundation Graduate Research Fellowship, Stanford Graduate Fellowship (Smith Fellow), and by the Stanford Cancer Biology Program T32 Grant (T32 CA009302). B.J.M was supported by a Stanford Graduate Fellowship (Lucille P. Markey Biomedical Research Fellowship). Y.S. received a fellowship from the Helmsley Charitable Trust. R.A.P.S. was partially funded by a graduate fellowship from the National Institute of Standards and Technology (NIST). L.L. was supported by a National Science Foundation Graduate Research Fellowship, Stanford Graduate Fellowship, and Stanford EDGE Fellowship. E.S. was supported by a National Science Scholarship, Agency for Science, Technology, and Research (A*STAR). H.C.W. was supported by a National Science Foundation Graduate Research Fellowship. N.M. was supported by a National Science Foundation Graduate Research Fellowship and Stanford Graduate Fellowship. This project was supported by a Stanford Cancer Institute Pancreatic Cancer Innovation Award (J.R.C.), a Stanford Coulter Foundation Translational Partnership Award (J.R.C.), a grant from NIH (CA082683) (T.H.), and awards from the Lustgarten Foundation (552873) (T.H.) and SU2C Pancreatic Cancer Dream Team (SU2C-AACR-DT-20-16) (T.H.). T.H. is a Frank and Else Schilling American Cancer Society Professor and the Renato Dulbecco Chair in Cancer Research.

## Author contributions

S.A.H. and J.R.C. conceived and designed the study. S.A.H. performed most of the experiments. B.J.M. performed binding analysis, LIFR mutant analysis, and assisted with xenograft studies. Y.S. conducted pilot in vivo KP$^{f/f}$CL pSTAT3 experiments, ultra-sensitive ELISA, and along with T.H., contributed expertise in LIF and PDAC biology as well as essential reagents. R.A.P.S. performed the LIFR structure modeling and analysis. L.L. assisted in LIFR library sorting and mutant analysis. C.F. performed Uncle melting and aggregation temperature analysis. E.S. optimized initial tumor spheroid assays and helped with xenograft studies. H.C.W. performed LIFR specificity studies. N.M. provided statistical and experimental design input as well as assistance in structure analysis. C.C. designed and tested heterodimeric one-arm LIFR variants. S.A.H., Y.S., T.H., and J.R.C. prepared the manuscript for submission. All authors provided intellectual input to the study and approved the final version of the manuscript.

## Competing interests

S.A.H. and J.R.C. are included as inventors on intellectual property related to the work described in this manuscript. J.R.C. is a co-founder and equity holder of xCella Bios-ciences, Trapeze Therapeutics, and Virsti Therapeutics, and an equity holder in Xyence Therapeutics and Aravive, which are developing protein therapeutics for oncology. No other authors have competing interests to declare.
