## [Peer Review File · Communications Biology]

Reviewers' comments:

Reviewer #1 (Remarks to the Author):

LIF is a pleiotropic cytokine that has recently been implicated in the pathogenesis of pancreatic cancer. Tumor cells and myofibroblasts found in the tumor stroma secrete this cytokine to promote tumor growth. In the cancer cells, this effect seems to be dependent on oncogenic KRAS, which is a genetic driver of most pancreatic adenocarcinomas. Mouse models have been used to demonstrate the therapeutic potential of inhibiting LIF in this setting, using both genetically and pharmacologically based approaches. However, additional and alternative anti-LIF therapies are of great interest given the potential clinical relevance.

The manuscript by Hunter and colleagues describes the generation of a ligand trap (also known as receptor decoy) that potently inhibits human LIF. Using an elegant yeast-based approach, the authors identify single amino acid variants in the LIFR that significantly improve the binding to LIF. This novel engineered ligand trap exhibits higher affinity than the wild type LIFR and is used to determine the therapeutic potential in different cellular and in vivo models.

Overall, I think that the manuscript is of great interest because it reports on a novel therapeutic modality to neutralize LIF. The authors have carefully characterized the biochemical properties of the receptor decoys and have identified novel variants that increase binding affinity. This is an excellent collaborative effort and I think it will be of great interest to a broad readership.

I only have two major comments about this manuscript.

1. The authors did a lot of work characterizing an initial proof-of-concept LIFR ligand trap and the results are "buried" in the supplementary materials. I would recommend presenting a main figure with some of these results. The data is informative and high-quality and should not be lost in the supplementary figures.

2. The excellent engineering work of the ligand trap is somehow eclipsed by an inadequate in vivo experiment. The xenograft using KP4-WT cells is underpowered. The low number of animals used in the experiment make it difficult to interpret the significance of the result. The xenograft using KP4-LIF cells has been done with a larger cohort of mice. However, by looking at the PBS control arm, it seems that many tumors did not engraft/grow as expected. Consistently, the tumors depicted in Supplementary Fig 8 underline this observation. Therefore, it is difficult to assess whether the effects of the treatment are real or a bias from randomization. Perhaps a greater number of cells should have been injected to increase engraftment. Also, why was the experiment stopped at such an early timepoint? It looks like the control tumors could have grown more and maximize the statistical difference. I understand that certain cell lines are difficult to grow in vivo, but perhaps a different model would be more useful. Alternatively, could the authors show the data as a % change from baseline (waterfall plot)?

These are some few additional suggestions that, in my opinion, would improve this manuscript.

- The authors do not discuss the potential implications of these ligand traps in fertility. Because LIF is highly expressed in placenta and knockout mice are infertile, I think it would be important to discuss it as a potential secondary effect if used as a therapy.

- In line 131, the authors refer to two different mouse strains as "healthy". I would rather use the term wild type or normal, given that these congenic strains generally exhibit certain conditions associated to their background.

- Could the authors speculate in the Discussion why the gp130 inhibition would be beneficial? There is already some discussion about the difference between the receptor decoy and the antibody in this regard but would be interesting to see whether the authors think there is a potential clinical benefit.
- Figure 2c is difficult to differentiate both colors. Figure 2b could depict the mutations used in the eLIFR decoy; this would help differentiate from the hLIFR.

Reviewer #2 (Remarks to the Author):

Here, the authors use yeast display to engineer a high affinity, soluble variant of leukemia inhibitory factor receptor (LIFR) to serve as a ligand trap for the LIF cytokine. Through a combination of in vitro and in vivo models, the authors demonstrate that the engineered LIFR (eLIFR) exhibits improved affinity relative to WT LIFR and that eLIFR can disrupt LIF signaling in cancer cells. The authors also test a variety of scaffolds that could serve as the basis for an eLIFR-based therapeutic. The authors conclude that trapping LIF with an Fc fusion of eLIFR (eLIFR-Fc) may be a viable anti-cancer therapeutic strategy.

Major Comments

1. The authors present a variety of different LIFR-Fc fusion proteins, settle on using an N-terminally fused eLIFR construct for a majority of their experiments, and conclude that this construct may be a viable therapeutic. The study would benefit from further characterization (expression levels, size exclusion chromatography profiles, and differential scanning fluorimetry) of these fusion proteins. Furthermore, comparisons of hLIFR versus eLIFR using these techniques would also help support the statement in the Discussion that the L218P mutation may stabilize the protein.
2. In the results, there are two instances where normalized binding is >1 relative to the control (D25 in in Figure 2F and CLCF-1 in Figure 4A). The authors briefly mention that D25 improves binding of LIF to gp130, but do not discuss why this might be the case. The authors should address potential causes for these trends.
3. In Figure 1G, eLIFR demonstrates greatly elevated LIF binding RFU at saturation than hLIFR, even though expression of the two seems comparable in Figure 1E. Similarly, in Supplementary Figure 5C, hLIFR-Fc seems to only permit 50% inhibition of LIF signaling, even at substantial excess. The authors should discuss potential reasons for these observations.
4. In Figure 3 the LIF concentration used in each panel varies nearly three orders of magnitude, from 20 pM to 12 nM. Although the diversity of the tumor microenvironment makes such an estimate difficult, can the authors comment on what the potential in vivo concentration of LIF may be in a tumor? Along these same lines, how representative is the KP4-LIF model of LIF levels that may be seen in actual tumors? This should be discussed.
5. In Supplementary Figure 6H and I, 25 nM hLIFR/eLIFR is used for KP4 spheroid experiments, whereas 20-fold more (500 nM) is used for PANC1 experiments. Why the dramatic difference in concentrations?
6. Overall, the Discussion is lacking in detail regarding trends seen in the in vivo models. For example, in several instances the authors observe bimodal distributions for eLIFR-Fc-treated mice in their in vivo models. What could be driving the differences in responses from tumor-to-tumor? A powerful additional experiment would be to perform immunohistochemistry or other means of quantifying LIF in

excised tumors to see if intratumoral LIF levels, normalized to tumor size, correlate with response. Perhaps this is driving the bimodal distribution? Along these same lines, in Supplementary Figure 8F, the D25 combination treatment did not produce a bimodal effect in tumor volume. What might cause this difference between eLIFR-Fc and D25? This should be discussed.

7. It is impressive that mLIFR binds human LIF with such high affinity. The authors mention that three of the mutations in eLIFR are found in mLIFR, but what might be other reasons for the high affinity binding observed with mLIFR? This should be discussed.

Minor Comments:

1. On page 11, line 236, there is a typo. "Even at LIF concentrations of only ten-fold excess..." should read "Even at LIFR concentrations of only ten-fold excess...".

2. In Supplementary Figure 4B the authors clearly spliced images of different gels together. Even if it is just the molecular weight marker, clear delineation between different gels (e.g. a black line) should be shown to demonstrate that different gels are being presented together.

COMMSBIO-20-1327A: “An engineered ligand trap is a potent inhibitor of LIF in pancreatic”

Response to reviewers

Reviewer #1 (Remarks to the Author):

LIF is a pleiotropic cytokine that has recently been implicated in the pathogenesis of pancreatic cancer. Tumor cells and myofibroblasts found in the tumor stroma secrete this cytokine to promote tumor growth. In the cancer cells, this effect seems to be dependent on oncogenic KRAS, which is a genetic driver of most pancreatic adenocarcinomas. Mouse models have been used to demonstrate the therapeutic potential of inhibiting LIF in this setting, using both genetically and pharmacologically based approaches. However, additional and alternative anti-LIF therapies are of great interest given the potential clinical relevance.

The manuscript by Hunter and colleagues describes the generation of a ligand trap (also known as receptor decoy) that potently inhibits human LIF. Using an elegant yeast-based approach, the authors identify single amino acid variants in the LIFR that significantly improve the binding to LIF. This novel engineered ligand trap exhibits higher affinity than the wild type LIFR and is used to determine the therapeutic potential in different cellular and in vivo models.

Overall, I think that the manuscript is of great interest because it reports on a novel therapeutic modality to neutralize LIF. The authors have carefully characterized the biochemical properties of the receptor decoys and have identified novel variants that increase binding affinity. This is an excellent collaborative effort and I think it will be of great interest to a broad readership.

I only have two major comments about this manuscript.

1. The authors did a lot of work characterizing an initial proof-of-concept LIFR ligand trap and the results are “buried” in the supplementary materials. I would recommend presenting a main figure with some of these results. The data is informative and high-quality and should not be lost in the supplementary figures.

○ **Response:** We thank the reviewer for this comment. We agree and have now made this in vivo work into a main figure, Fig. 1

2. The excellent engineering work of the ligand trap is somehow eclipsed by an inadequate in vivo experiment. The xenograft using KP4-WT cells is underpowered. The low number of animals used in the experiment make it difficult to interpret the significance of the result. The xenograft using KP4-LIF cells has been done with a larger cohort of mice. However, by looking at the PBS control arm, it seems that many tumors did not engraft/grow as expected. Consistently, the tumors depicted in Supplementary Fig 8 underline this observation. Therefore, it is difficult to assess whether the effects of the treatment are real or a bias from randomization. Perhaps a greater number of cells should have been injected to increase engraftment. Also, why was the experiment stopped at such an early timepoint? It looks like the control tumors could have grown more and maximize the statistical difference. I understand that certain cell lines are difficult to grow in vivo, but perhaps a different model would be more useful. Alternatively, could the authors show the data as a % change from baseline (waterfall plot)?

○ **Response:** We appreciate that this model is suboptimal, and thank Reviewer 1 for their suggestions to improve this model and make it more reliable. In terms of early timepoint termination, we unfortunately needed to stop the initial experiment early due to COVID constraints and lab shutdown. Thus, these experiments were the focus of our revision efforts, given the editor’s emphasis that this is the main area that should be addressed. In our initial experiments, we had tried to increase the number of mice used in each treatment arm of the KP4-LIF xenograft to minimize the effects of bias or randomization and address the heterogeneity of this model, as pointed out by the reviewer.

○ We set out to further address these concerns by following the reviewer’s advice:

- We injected 5e6 cells/mouse (up from 2e6 cells/mouse)
 - Waited longer to bin mice to allow tumors to become established (~2.5 weeks, up from ~1 week)
 - Only kept those mice in the study with large tumors that were growing well (top ~50% of mice)
 - Kept the experiment going as long as possible, until the tumors reached the euthanasia criteria established by our protocol.
- We used this modified protocol design for additional KP4-LIF and KP4-WT studies, as we agreed with the reviewer that the KP4-WT study was underpowered and unoptimized, making the initial results potentially unreliable.

KP4-LIF:

While there was again a significant benefit in treatment with the combination of eLIFR-Fc and gemcitabine in this model, we still observed heterogeneity of size/engraftment in the untreated PBS control, undermining confidence in the results. Because of this issue, and concerns raised initially by both reviewers, **we have moved away from this model** for the revised manuscript.

KP4 (wild-type, WT):

We are appreciative that Reviewer 1 requested that we repeat the KP4 xenograft study, as this optimized version of the study showed a significant abrogation in tumor growth with eLIFR-Fc treatment alone, compared to PBS. We were able to achieve this result compared to the previous KP4 study in the initial paper for several reasons:

- (1) This study was optimized, as per the suggestions of the reviewer, to have more robust tumors.
- (2) As also pointed out by the reviewer, the previous study was under-powered and did not have enough mice to make a meaningful conclusion. Repeating the study with more mice revealed that there was a significant effect in treating with eLIFR-Fc vs PBS.
- (3) We had initially thought that KP4 cells did not produce high enough levels of LIF to be used as a xenograft, hence the rationale for making KP4-LIF cells. However, we have now been able to measure secreted LIF levels from the KP4 cells using an ultra-sensitive ELISA (data on next page).

KP4 Repeat Xenograft Model. (a) Treatment of human KP4 PDAC xenograft model with PBS or eLIFR-Fc (10 mg/kg, 2x/week). Dosing schedule indicated by arrows. $P=0.05$; $*P=0.02$; $**P=0.009$ by two-way ANOVA with Sidak's multiple comparisons test. Data are the mean \pm SEM ($n = 7$). (b) Tumor measurements from individual mice in KP4 study. Dosing schedule indicated by arrows. (c) Weights of excised KP4 treated tumors. $*P=0.03$ versus PBS treated tumors by two-tailed unpaired Student's t -test. Data are the mean \pm SD ($n = 7$). (d) Volumes of excised KP4 treated tumors. $**P=0.006$ versus PBS treated tumors by two-tailed unpaired Student's t -test. Data are the mean \pm SD ($n = 7$).

The figure at the left shows the LIF concentration measured in conditioned media collected from KP4 cells, quantified by ultra-sensitive ELISA, minus the baseline [LIF] in media alone. Data are the mean \pm SD ($n = 5$). Thus, with this new measurement technique, we were able to confirm that KP4 cells do secrete LIF, providing confidence that they should be susceptible to LIF inhibition as an in vivo xenograft model.

Based on the above findings, **we replaced the KP4-LIF data in the revised manuscript with the new KP4 xenograft data.** We believe that this KP4 optimized xenograft is more streamlined with the rest of the data provided, which does not use the transduced KP4-LIF cell line, and does not suffer from the same concerns over effects from randomization or bias. We also observed single-agent efficacy with the treatment of eLIFR-Fc alone in the KP4 model.

Reviewer #1: These are some few additional suggestions that, in my opinion, would improve this manuscript.

- The authors do not discuss the potential implications of these ligand traps in fertility. Because LIF is highly expressed in placenta and knockout mice are infertile, I think it would be important to discuss it as a potential secondary effect if used as a therapy.

- o **Response:** LIF's role in fertility is well known, and we have now included discussion on this topic.
 - *Text from Results (Lines 124 – 128):* "LIF inhibition is expected to be well tolerated, as LIF expression is relatively low in adult mice and humans, while mutations or deletions of the LIF gene does not lead to negative health consequences, save for women presenting lower conception rates likely due to the fact that LIF is required for embryo implantation.^{3,35,36}

- In line 131, the authors refer to two different mouse strains as “healthy”. I would rather use the term wild type or normal, given that these congenic strains generally exhibit certain conditions associated to their background.

o **Response:** We agree that “healthy” is not the best word choice. The use was meant to clarify that these mice do not have cancer (that we know of). We changed this to a better and more specific wording: “non-tumor bearing wild-type mice.”

▪ *Text from Results (Lines 130 – 131):* “We sought to determine the effects of treatment with a LIFR-based ligand trap, dosing two strains of non-tumor bearing wild-type mice (FVB and Black/6) 3 times per week with phosphate buffered saline or 20 mg/kg mLIFR-Fc for 1 month.”

- Could the authors speculate in the Discussion why the gp130 inhibition would be beneficial? There is already some discussion about the difference between the receptor decoy and the antibody in this regard but would be interesting to see whether the authors think there is a potential clinical benefit.

o **Response:** We have added a brief sentence in the discussion, taking care not to be overly speculative.

▪ *Text from Discussion (Lines 387 – 390):* “Such dual-specific competition of both LIFR and gp130 could potentially contribute to eLIFR-Fc inhibition, for example, if LIF binding to cell surface gp130 in the absence of LIFR binding still increases the local concentration of soluble factor.”

- Figure 2c is difficult to differentiate both colors. Figure 2b could depict the mutations used in the eLIFR decoy; this would help differentiate from the hLIFR.

o **Response:** We have made changes throughout to make eLIFR and hLIFR more distinct by including cartoons of each molecule. We have altered the colors throughout, including in 2c (now 3c) and added a cartoon of hLIFR-Fc to 2b (now 3b) to make the distinction of the altered Ig-like domain clearer.

Fig 3b:

Fig 3c:

Reviewer #2 (Remarks to the Author):

Here, the authors use yeast display to engineer a high affinity, soluble variant of leukemia inhibitory factor receptor (LIFR) to serve as a ligand trap for the LIF cytokine. Through a combination of in vitro and in vivo models, the authors demonstrate that the engineered LIFR (eLIFR) exhibits improved affinity relative to WT LIFR and that eLIFR can disrupt LIF signaling in cancer cells. The authors also test a variety of scaffolds that could serve as the basis for an eLIFR-based therapeutic. The authors conclude that trapping LIF with an Fc fusion of eLIFR (eLIFR-Fc) may be a viable anti-cancer therapeutic strategy.

Major Comments

1. The authors present a variety of different LIFR-Fc fusion proteins, settle on using an N-terminally fused eLIFR construct for a majority of their experiments, and conclude that this construct may be a viable therapeutic. The study would benefit from further characterization (expression levels, size exclusion chromatography profiles, and differential scanning fluorimetry) of these fusion proteins. Furthermore, comparisons of hLIFR versus eLIFR using these techniques would also help support the statement in the Discussion that the L218P mutation may stabilize the protein.

- Response:** We performed comparisons between eLIFR-Fc and hLIFR-Fc and have amended Supplemental Figure 3 d & e (respectively) to include a representative size exclusion chromatography (SEC) trace of each protein (demonstrating overall good purity, with a slight shoulder for hLIFR-Fc) as well as melting temperature and aggregation analysis using fluorescence, SLS, and DLS (performed using an Uncle instrument, Unchained Labs). These data reveal very similar properties between hLIFR-Fc and eLIFR-Fc, indicating that the improved affinity or functional performance of eLIFR is likely not due to improved thermal stability. These data also show that LIFR-Fc (both wild-type and engineered) are not prone to aggregation, even at high temperatures (as revealed by SLS and DLS data), which is a favorable quality for therapeutic development.

SEC (Supp. Fig. 3d):

Fluorescence, SLS, and DLS (Supp. Fig. 3e):

Thermal melt:

	T _m 1 (°C)	T _m 2 (°C)
hLIFR-Fc	50.5 ± 0.4	76.8 ± 0.3
eLIFR-Fc	50.7 ± 0.4	81.8 ± 0.3
IgG	71.4 ± 0.3	-

DLS Average Particle Diameter:

	15 °C	95 °C
hLIFR-Fc	68.0 nm	39.7 nm
eLIFR-Fc	57.6 nm	45.1 nm
IgG	11.0 nm	>1000 nm

- One interesting result of these assays was the observation that LIFR-Fc behaves in a very differently than a traditional antibody. In parallel we characterized a control IgG (dinutuximab), purified in the same manner as LIFR-Fc. The control IgG displayed a single T_m of 71.4 °C, which was associated with prominent aggregation. DLS measurements at 15 °C revealed an average particle diameter of 11.0 nm, but a particle diameter >1000 nm at 95 °C, indicating large aggregates. This is a typical profile for a

monoclonal antibody, but is very different from the properties observed for either LIFR-Fc variant.

- *Text added on these results in the Results (Lines 212 – 217):* “Purified eLIFR-Fc and hLIFR-Fc were assessed by analytical size exclusion chromatography (**Supplementary Fig. 3d**). Both purified proteins displayed near identical melting temperatures and a strikingly limited propensity for aggregation, especially when compared to a control IgG, as measured using intrinsic fluorescence, static light scattering (SLS), and dynamic light scattering (DLS) on an Uncle instrument (**Supplementary Fig. 3e**).”
- *And Discussion (Lines 331 – 337):* “These changes do not result in improved thermal stability (**Supplementary Fig. 3e**), indicating that the improved properties demonstrated by eLIFR-Fc is likely not the result of improved stability over hLIFR-Fc. Both LIFR-Fc proteins displayed minimal aggregation (**Supplementary Fig. 3d,e**), even at high temperatures, a favorable quality for therapeutic development, and in stark contrast to a control IgG, which formed large aggregates at higher temperatures (**Supplementary Fig. 3e**).”
 - Providing similar biophysical data for every LIFR-Fc fusion construct is not feasible at this time, and perhaps tangential as we did not move forward with these variants, since they do not bind to LIF with as high affinity as eLIFR-Fc. Because it was speculative, and not supported by the Uncle data, we have removed the statement from the Discussion of the potential benefits of L2 18P. Expression of the fusion proteins were not optimized; as an academic lab, our goal was to obtain enough protein for the studies performed in the paper, and thus discussion of expression yields would be misleading in this context.

2. In the results, there are two instances where normalized binding is >1 relative to the control (D25 in in Figure 2F and CLCF-1 in Figure 4A). The authors briefly mention that D25 improves binding of LIF to gp130, but do not discuss why this might be the case. The authors should address potential causes for these trends.

- **Response:** We have addressed these points in the Figure legends that they correspond to:
 - Figure 2f (now Fig. 3f): There are two potential mechanisms for increased binding of LIF to gp130: (1) D25 binding locks LIF in a favorable conformation for gp130 binding or (2) being bivalent, a complex formed of YSD-gp130-LIF-D25-LIF is possible. This would double the amount of LIF bound, which would readily explain the ~2 fold increase in binding. This is also one rationale for why One-Arm eLIFR-Fc is a better apparent gp130 competitor than bivalent eLIFR-Fc. Without the second LIFR fragment, the only complex that can be formed is YSD-gp130-LIF-One Arm eLIFR-Fc. Thus, the inhibition signal observed with eLIFR-Fc is likely underestimated.
 - *Text in Fig. 3f legend:* “**(f)** Both hLIFR-Fc and eLIFR-Fc (bivalent and one-arm) compete LIF away from WT gp130, but the D25 mAb does not and appears to increase binding, perhaps due to complex stabilization or more avid LIF binding.”
 - Figure 4a (now Supplementary Fig. 6d): The likely cause is mild agonism. It should be noted that the signal from both CTF-1 and CLCF-1 is very weak in HeLa cells (see Supplementary Fig. 6c), so a small increase in signal can look larger than it is when normalized, as is the case here. Further, CLCF-1 specifically is rather unstable, meaning that in this context it might be aided by soluble LIFR, which binds weakly, in forming a complex on the surface of cells, allowing for cell-surface LIFR to replace soluble LIFR and drive signaling. Given that LIFR is at such a high concentration, we doubt that this would be able to occur in vivo, and think that this is more of an artifact of this particular assay, and maintain the point that LIFR-Fc does not inhibit signaling for any of these cytokines.

- *Text in Supp. Fig. 6d legend: “(d) High concentrations of eLIFR-Fc or hLIFR-Fc minimally inhibit or possibly promote mild agonism of CTF-1 or CLCF-1 signaling in HeLa STAT3 cells as compared to LIF.”*

3. In Figure 1G, eLIFR demonstrates greatly elevated LIF binding RFU at saturation than hLIFR, even though expression of the two seems comparable in Figure 1E. Similarly, in Supplementary Figure 5C, hLIFR-Fc seems to only permit 50% inhibition of LIF signaling, even at substantial excess. The authors should discuss potential reasons for these observations.

○ **Response:** We have addressed these points in the Figure legends that they correspond to:

- Figure 1g (now Fig. 2g): Our lab almost always observes a higher maximum binding signal with higher affinity proteins in yeast-surface display assays. We interpret this as a slower off rate, which we show to be the case with eLIFR vs hLIFR in Fig. 3c. During the staining, washing, and flow steps, LIF is dissociating from hLIFR at a greater rate than eLIFR, which reduces the overall signal. Thus, in our eyes, both a leftward and upward shift of the binding curve in YSD binding assays are good indications of improved affinity.
 - *Text from Fig. 2g legend: “(g) Yeast-displayed eLIFR binds LIF-His with a higher affinity than WT hLIFR. Higher maximum binding signal from eLIFR likely indicates a slower off-rate of binding.”*
- Supplementary Figure 5c (now Supplementary Fig. 4c): It is true that hLIFR-Fc seems ineffective even at higher doses. In some ways this should be expected, as hLIFR-Fc has no competitive advantage over hLIFR on the surface of cells. In this assay, we used a 40-fold excess (20 nM LIFR-Fc vs 0.5 nM LIF). It is worth noting that in the IC₅₀ experiment (Fig. 3a), a 50% reduction in signal was not observed until a 90-fold excess (1.8 nM LIFR-Fc vs 0.02 nM LIF). These data suggest that hLIFR-Fc needs to be in excess of LIF by >100-fold to achieve robust inhibition, vastly higher than the concentration of eLIFR-Fc required (<2-fold).
 - *Text from Supp. Fig. 4c legend: “(c) eLIFR-Fc nearly completely silences LIF signaling, while hLIFR-Fc, with no competitive LIF binding advantage, is ineffective at silencing LIF signaling, even at concentrations 40-fold in excess.”*

4. In Figure 3 the LIF concentration used in each panel varies nearly three orders of magnitude, from 20 pM to 12 nM. Although the diversity of the tumor microenvironment makes such an estimate difficult, can the authors comment on what the potential in vivo concentration of LIF may be in a tumor? Along these same lines, how representative is the KP4-LIF model of LIF levels that may be seen in actual tumors? This should be discussed.

○ **Response:**

- Referring to data from a recent publication of our co-authors, Yu Shi and Tony Hunter (Shi, et. al., *Nature* 2019), LIF was measured in human PDAC tumors (see Shi, et. al. Fig. 5, Extended Data Fig. 10), at ~1 ng/mg protein in tumor lysate. While pancreatic lesions can vary in size, a back of the envelope calculation gives that for an 8 g tumor (~2 cm³) in which ~2% of the mass is derived from protein, LIF can be expected at a concentration of ~1 pM in this 8 mL lesion. As mentioned, this is likely an under-estimate and could vary widely by tumor, but demonstrates that testing for efficacy in the pM – nM range is reasonable.
- We reference this calculation in the methods section describing the HeLa assay, but as it is a rough calculation, do not want to put too much stock in it:
 - *Text from Materials and Methods (Lines 691 – 695): “Proteins were added to serum-starved HeLa cells for 4 h at 37 °C. For IC₅₀ analysis, eLIFR-Fc or hLIFR-Fc were*

serially diluted (eLIFR: 30 nM to 1.5 pM; hLIFR: 300 nM to 15 pM) in HeLa assay medium with a constant concentration of 20 pM LIF, reflective of the possible concentration of LIF in the pancreatic cancer microenvironment.³

- We measured LIF levels by ultra-sensitive ELISA for the KP4 tumors extracted during the study. These data are not in the revised submission itself, but to address this reviewer question, we found that on average in PBS treated KP4 tumors, LIF was present at a concentration of ~75 pg/mg of protein lysate. This concentration is lower than that estimated for patient samples, but given the relatively high level of LIF secretion observed in these cells (see above, and data repeated below), this is likely an underestimation of LIF levels, as LIF is a soluble factor, and when secreted will be diluted in the surrounding serum, and hence not effectively captured in tumor lysates. As we have found with all of the traps engineered in our lab, it is not important to just inhibit the factor from the tumor, but also the serum.

[LIF] measured in tumors treated with PBS:

[LIF] secreted into conditioned media by KP4 cells in vitro:

5. In Supplementary Figure 6H and I, 25 nM hLIFR/eLIFR is used for KP4 spheroid experiments, whereas 20-fold more (500 nM) is used for PANC1 experiments. Why the dramatic difference in concentrations?

Response: We did test lower concentrations of LIFR-Fc in PANC1, and eLIFR-Fc did somewhat reduce sphere formation at 25 nM and 150 nM (as opposed to hLIFR-Fc, which was ineffective), but it was not nearly as potent at lower concentrations. We have now included all concentrations in Fig. 5i:

Supplementary Fig. 5(i) Quantification of total sphere area in PANC1 cells after 2 weeks of growth, treated with 25 nM, 150 nM, or 500 nM eLIFR-Fc or hLIFR-Fc. [LIF] = 12 nM. ** $P = 0.0025$, *** $P = 0.0006$ compared to “plus LIF” or between LIFR-Fc treatments, as indicated, by two-tailed unpaired Student’s t -test. Data are the mean total sphere area \pm SD ($n = 6$).

- We found that PANC1 cells form spheres much less robustly than KP4 cells as evidenced by the small dynamic range. These cells are hence not as optimal for use in this assay, thus are presented only as a secondary cell line.

6. Overall, the Discussion is lacking in detail regarding trends seen in the in vivo models. For example, in several instances the authors observe bimodal distributions for eLIFR-Fc-treated mice in their in vivo models. What could be driving the differences in responses from tumor-to-tumor? A powerful additional experiment would be to perform immunohistochemistry or other means of quantifying LIF in excised tumors to see if intratumoral LIF levels, normalized to tumor size, correlate with response. Perhaps this is driving the bimodal distribution? Along these same lines, in Supplementary Figure 8F, the D25 combination treatment did not produce a bimodal effect in tumor volume. What might cause this difference between eLIFR-Fc and D25? This should be discussed.

- **Response:** As discussed in the response to Reviewer 1, we agree that the seemingly bimodal distribution observed in the KP4-LIF xenograft model is unusual. Given that this trend was also apparent in the PBS-treated group, it appears likely to be an effect driven by the cell line, and not treatment. KP4-LIF is a heterogenous cell line (not derived from a single clonal population after transduction).
- As discussed above, due to the unusual nature of this xenograft model, and our success in optimizing the KP4 xenograft model, we are replacing the KP4-LIF data with the new KP4 data in the revised manuscript.
- We analyzed LIF levels in tumor lysates from the KP4 WT study by ultra-sensitive ELISA, summarized below. There was no correlation observed between tumor size and LIF expression in the PBS treated group, despite a range of LIF protein levels. There was a moderate correlation between LIF levels and tumor size in the eLIFR-Fc treated group ($R^2=0.70$), however, not enough data to draw a strong conclusion.

LIF levels in KP4 WT tumor lysates. (a) LIF levels quantified by ultrasensitive ELISA from lysates of KP4 WT tumors treated with PBS or eLIFR-Fc. Data are the mean \pm SD ($n = 7$). **(b)** Correlation of LIF levels quantified by ultrasensitive ELISA from lysates of KP4 tumors with excised tumor weight. Treatment groups are shown as different colors (PBS: black; Purple: eLIFR-Fc). **(c)** Correlation between LIF levels and excised tumor weight in individual treatment groups, as described in **b**.

7. It is impressive that mLIFR binds human LIF with such high affinity. The authors mention that three of the mutations in eLIFR are found in mLIFR, but what might be other reasons for the high affinity binding observed with mLIFR? This should be discussed.

- **Response:** This is a challenging question, and one that we have explored with computational modeling, but which does not have a readily apparent answer. In all, it seems to be the contributions of many residues and regions, though it is beyond the scope of this work to profile them.
- We allude to one possible explanation/solution through identifying binding interactions present in other domains of LIFR besides the Ig-like domain in the Discussion:
 - *Text from Discussion (Lines 348 – 351):* “Further improvements in affinity and species cross-reactivity may thus be possible through insertion of other regions of mLIFR that contact LIF (such as the CBM II) into the eLIFR scaffold, however, additional changes may result in increased immunogenicity.”

Minor Comments:

1. On page 11, line 236, there is a typo. “Even at LIF concentrations of only ten-fold excess...” should read “Even at LIFR concentrations of only ten-fold excess...”.

- **Response:** Good catch! This change has been made.

2. In Supplementary Figure 4B the authors clearly spliced images of different gels together. Even if it is just the molecular weight marker, clear delineation between different gels (e.g. a black line) should be shown to demonstrate that different gels are b

- **Response:** Thank you! It was not our intent to have these gels seem spliced together – lanes are placed together for clarity. We have put lines between the gels to indicate that they have been spliced together to avoid confusion that the lanes were run side by side. The legend now makes clear that the lanes were taken from the same gel and have now been placed side-by-side for clarity with lines indicating lanes that were not adjacent on the original gel.

REVIEWERS' COMMENTS:

Reviewer #1 (Remarks to the Author):

The authors have addressed my concerns.
Congratulations on this beautiful paper!

Reviewer #2 (Remarks to the Author):

I thank the authors for their thorough and thoughtful responses to my initial concerns. The new data characterizing eLIFR-Fc strengthens the manuscript, as does the use of a new mouse model to evaluate the efficacy of eLIFR-Fc in vivo. I have no further concerns, aside from noting that in Supplementary Figure 6G, "Excised" in the y-axis label is misspelled.

COMMSBIO-20-1327A: "An engineered ligand trap is a potent inhibitor of LIF in pancreatic"

Referee expertise:

Referee #1: pancreatic cancer, KRAS driven signaling

Referee #2: recombinant antibody, pharmaceutical chemistry

Reviewer #1 (Remarks to the Author):

LIF is a pleiotropic cytokine that has recently been implicated in the pathogenesis of pancreatic cancer. Tumor cells and myofibroblasts found in the tumor stroma secrete this cytokine to promote tumor growth. In the cancer cells, this effect seems to be dependent on oncogenic KRAS, which is a genetic driver of most pancreatic adenocarcinomas. Mouse models have been used to demonstrate the therapeutic potential of inhibiting LIF in this setting, using both genetically and pharmacologically based approaches. However, additional and alternative anti-LIF therapies are of great interest given the potential clinical relevance.

The manuscript by Hunter and colleagues describes the generation of a ligand trap (also known as receptor decoy) that potently inhibits human LIF. Using an elegant yeast-based approach, the authors identify single amino acid variants in the LIFR that significantly improve the binding to LIF. This novel engineered ligand trap exhibits higher affinity than the wild type LIFR and is used to determine the therapeutic potential in different cellular and in vivo models.

Overall, I think that the manuscript is of great interest because it reports on a novel therapeutic modality to neutralize LIF. The authors have carefully characterized the biochemical properties of the receptor decoys and have identified novel variants that increase binding affinity. This is an excellent collaborative effort and I think it will be of great interest to a broad readership.

I only have two major comments about this manuscript.

1. The authors did a lot of work characterizing an initial proof-of-concept LIFR ligand trap and the results are "buried" in the supplementary materials. I would recommend presenting a main figure with some of these results. The data is informative and high-quality and should not be lost in the supplementary figures.

○ **Response:** We thank the reviewer for this comment. We agree and have now made this in vivo work into a main figure, Fig. 1

2. The excellent engineering work of the ligand trap is somehow eclipsed by an inadequate in vivo experiment. The xenograft using KP4-WT cells is underpowered. The low number of animals used in the experiment make it difficult to interpret the significance of the result. The xenograft using KP4-LIF cells has been done with a larger cohort of mice. However, by looking at the PBS control arm, it seems that many tumors did not engraft/grow as expected. Consistently, the tumors depicted in Supplementary Fig 8 underline this observation.

Therefore, it is difficult to assess whether the effects of the treatment are real or a bias from randomization. Perhaps a greater number of cells should have been injected to increase engraftment. Also, why was the experiment stopped at such an early timepoint? It looks like the control tumors could have grown more and maximize the statistical difference. I understand that certain cell lines are difficult to grow in vivo, but perhaps a different model would be more useful. Alternatively, could the authors show the data as a % change from baseline (waterfall plot)?

- **Response:** We appreciate that this model is suboptimal, and thank Reviewer 1 for their suggestions to improve this model and make it more reliable. In terms of early timepoint termination, we unfortunately needed to stop the initial experiment early due to COVID constraints and lab shutdown. Thus, these experiments were the focus of our revision efforts, given the editor's emphasis that this is the main area that should be addressed. In our initial experiments, we had tried to increase the number of mice used in each treatment arm of the KP4-LIF xenograft to minimize the effects of bias or randomization and address the heterogeneity of this model, as pointed out by the reviewer.
 - We set out to further address these concerns by following the reviewer's advice:
 - We injected 5e6 cells/mouse (up from 2e6 cells/mouse)
 - Waited longer to bin mice to allow tumors to become established (~2.5 weeks, up from ~1 week)
 - Only kept those mice in the study with large tumors that were growing well (top ~50% of mice)
 - Kept the experiment going as long as possible, until the tumors reached the euthanasia criteria established by our protocol.
 - We used this modified protocol design for additional KP4-LIF and KP4-WT studies, as we agreed with the reviewer that the KP4-WT study was underpowered and unoptimized, making the initial results potentially unreliable.

KP4-LIF:

While there was again a significant benefit in treatment with the combination of eLIFR-Fc and gemcitabine in this model, we still observed heterogeneity of size/engraftment in the untreated PBS control, undermining confidence in the results. Because of this issue, and concerns raised initially by both reviewers, **we have moved away from this model** for the revised manuscript.

KP4 (wild-type, WT):

We are appreciative that Reviewer 1 requested that we repeat the KP4 xenograft study, as this optimized version of the study showed a significant abrogation in tumor growth with eLIFR-Fc treatment alone, compared to PBS. We were able to achieve this result compared to the previous KP4 study in the initial paper for several reasons:

- (1) This study was optimized, as per the suggestions of the reviewer, to have more robust tumors.
- (2) As also pointed out by the reviewer, the previous study was under-powered and did not have enough mice to make a meaningful conclusion. Repeating the study with more mice revealed that there was a significant effect in treating with eLIFR-Fc vs PBS.
- (3) We had initially thought that KP4 cells did not produce high enough levels of LIF to be used as a xenograft, hence the rationale for making KP4-LIF cells. However, we have now been able to measure secreted LIF levels from the KP4 cells using an ultra-sensitive ELISA (data on next page).

KP4 Repeat Xenograft Model. (a) Treatment of human KP4 PDAC xenograft model with PBS or eLIFR-Fc (10 mg/kg, 2x/week). Dosing schedule indicated by arrows. $P=0.05$; $*P=0.02$; $**P=0.009$ by two-way ANOVA with Sidak's multiple comparisons test. Data are the mean \pm SEM ($n = 7$). (b) Tumor measurements from individual mice in KP4 study. Dosing schedule indicated by arrows. (c) Weights of excised KP4 treated tumors. $*P=0.03$ versus PBS treated tumors by two-tailed unpaired Student's t -test. Data are the mean \pm SD ($n = 7$). (d) Volumes of excised KP4 treated tumors. $**P=0.006$ versus PBS treated tumors by two-tailed unpaired Student's t -test. Data are the mean \pm SD ($n = 7$).

The figure at the left shows the LIF concentration measured in conditioned media collected from KP4 cells, quantified by ultra-sensitive ELISA, minus the baseline [LIF] in media alone. Data are the mean \pm SD ($n = 5$). Thus, with this new measurement technique, we were able to confirm that KP4 cells do secrete LIF, providing confidence that they should be susceptible to LIF inhibition as an in vivo xenograft model.

Based on the above findings, **we replaced the KP4-LIF data in the revised manuscript with the new KP4 xenograft data.** We believe that this KP4 optimized xenograft is more streamlined with the rest of the data provided, which does not use the transduced KP4-LIF cell line, and does not suffer from the same concerns over effects from randomization or bias. We also observed single-agent efficacy with the treatment of eLIFR-Fc alone in the KP4 model.

Reviewer #1: These are some few additional suggestions that, in my opinion, would improve this manuscript.

- The authors do not discuss the potential implications of these ligand traps in fertility. Because LIF is highly expressed in placenta and knockout mice are infertile, I think it would be important to discuss it as a potential secondary effect if used as a therapy.

- o **Response:** LIF's role in fertility is well known, and we have now included discussion on this topic.
 - *Text from Results (Lines 124 – 128):* "LIF inhibition is expected to be well tolerated, as LIF expression is relatively low in adult mice and humans, while mutations or deletions of the LIF gene does not lead to negative health consequences, save for women presenting lower conception rates likely due to the fact that LIF is required for embryo implantation.^{3,35,36}

- In line 131, the authors refer to two different mouse strains as “healthy”. I would rather use the term wild type or normal, given that these congenic strains generally exhibit certain conditions associated to their background.

o **Response:** We agree that “healthy” is not the best word choice. The use was meant to clarify that these mice do not have cancer (that we know of). We changed this to a better and more specific wording: “non-tumor bearing wild-type mice.”

▪ *Text from Results (Lines 130 – 131):* “We sought to determine the effects of treatment with a LIFR-based ligand trap, dosing two strains of non-tumor bearing wild-type mice (FVB and Black/6) 3 times per week with phosphate buffered saline or 20 mg/kg mLIFR-Fc for 1 month.”

- Could the authors speculate in the Discussion why the gp130 inhibition would be beneficial? There is already some discussion about the difference between the receptor decoy and the antibody in this regard but would be interesting to see whether the authors think there is a potential clinical benefit.

o **Response:** We have added a brief sentence in the discussion, taking care not to be overly speculative.

▪ *Text from Discussion (Lines 387 – 390):* “Such dual-specific competition of both LIFR and gp130 could potentially contribute to eLIFR-Fc inhibition, for example, if LIF binding to cell surface gp130 in the absence of LIFR binding still increases the local concentration of soluble factor.”

- Figure 2c is difficult to differentiate both colors. Figure 2b could depict the mutations used in the eLIFR decoy; this would help differentiate from the hLIFR.

o **Response:** We have made changes throughout to make eLIFR and hLIFR more distinct by including cartoons of each molecule. We have altered the colors throughout, including in 2c (now 3c) and added a cartoon of hLIFR-Fc to 2b (now 3b) to make the distinction of the altered Ig-like domain clearer.

Fig 3b:

Fig 3c:

Reviewer #2 (Remarks to the Author):

Here, the authors use yeast display to engineer a high affinity, soluble variant of leukemia inhibitory factor receptor (LIFR) to serve as a ligand trap for the LIF cytokine. Through a combination of in vitro and in vivo models, the authors demonstrate that the engineered LIFR (eLIFR) exhibits improved affinity relative to WT LIFR and that eLIFR can disrupt LIF signaling in cancer cells. The authors also test a variety of scaffolds that could serve as the basis for an eLIFR-based therapeutic. The authors conclude that trapping LIF with an Fc fusion of eLIFR (eLIFR-Fc) may be a viable anti-cancer therapeutic strategy.

Major Comments

1. The authors present a variety of different LIFR-Fc fusion proteins, settle on using an N-terminally fused eLIFR construct for a majority of their experiments, and conclude that this construct may be a viable therapeutic. The study would benefit from further characterization (expression levels, size exclusion chromatography profiles, and differential scanning fluorimetry) of these fusion proteins. Furthermore, comparisons of hLIFR versus eLIFR using these techniques would also help support the statement in the Discussion that the L218P mutation may stabilize the protein.

- **Response:** We performed comparisons between eLIFR-Fc and hLIFR-Fc and have amended Supplemental Figure 3 d & e (respectively) to include a representative size exclusion chromatography (SEC) trace of each protein (demonstrating overall good purity, with a slight shoulder for hLIFR-Fc) as well as melting temperature and aggregation analysis using fluorescence, SLS, and DLS (performed using an Uncle instrument, Unchained Labs). These data reveal very similar properties between hLIFR-Fc and eLIFR-Fc, indicating that the improved affinity or functional performance of eLIFR is likely not due to improved thermal stability. These data also show that LIFR-Fc (both wild-type and engineered) are not prone to aggregation, even at high temperatures (as revealed by SLS and DLS data), which is a favorable quality for therapeutic development.

SEC (Supp. Fig. 3d):

Fluorescence, SLS, and DLS (Supp. Fig. 3e):

Thermal melt:

	T _m 1 (°C)	T _m 2 (°C)
hLIFR-Fc	50.5 ± 0.4	76.8 ± 0.3
eLIFR-Fc	50.7 ± 0.4	81.8 ± 0.3
IgG	71.4 ± 0.3	-

DLS Average Particle Diameter:

	15 °C	95 °C
hLIFR-Fc	68.0 nm	39.7 nm
eLIFR-Fc	57.6 nm	45.1 nm
IgG	11.0 nm	>1000 nm

- One interesting result of these assays was the observation that LIFR-Fc behaves in a very differently than a traditional antibody. In parallel we characterized a control IgG (dinutuximab), purified in the same manner as LIFR-Fc. The control IgG displayed a single T_m of 71.4 °C, which was associated with prominent aggregation. DLS measurements at 15 °C revealed an average particle diameter of 11.0 nm, but a particle diameter >1000 nm at 95 °C, indicating large aggregates. This is a typical profile for a monoclonal antibody, but is very different from the properties observed for either LIFR-Fc variant.

- *Text added on these results in the Results (Lines 212 – 217):* “Purified eLIFR-Fc and hLIFR-Fc were assessed by analytical size exclusion chromatography (**Supplementary Fig. 3d**). Both purified proteins displayed near identical melting temperatures and a strikingly limited propensity for aggregation, especially when compared to a control IgG, as measured using intrinsic fluorescence, static light scattering (SLS), and dynamic light scattering (DLS) on an Uncle instrument (**Supplementary Fig. 3e**).”
- *And Discussion (Lines 331 – 337):* “These changes do not result in improved thermal stability (**Supplementary Fig. 3e**), indicating that the improved properties demonstrated by eLIFR-Fc is likely not the result of improved stability over hLIFR-Fc. Both LIFR-Fc proteins displayed minimal aggregation (**Supplementary Fig. 3d,e**), even at high temperatures, a favorable quality for therapeutic development, and in stark contrast to a control IgG, which formed large aggregates at higher temperatures (**Supplementary Fig. 3e**).”
 - Providing similar biophysical data for every LIFR-Fc fusion construct is not feasible at this time, and perhaps tangential as we did not move forward with these variants, since they do not bind to LIF with as high affinity as eLIFR-Fc. Because it was speculative, and not supported by the Uncle data, we have removed the statement from the Discussion of the potential benefits of L2 18P. Expression of the fusion proteins were not optimized; as an academic lab, our goal was to obtain enough protein for the studies performed in the paper, and thus discussion of expression yields would be misleading in this context.

2. In the results, there are two instances where normalized binding is >1 relative to the control (D25 in in Figure 2F and CLCF-1 in Figure 4A). The authors briefly mention that D25 improves binding of LIF to gp130, but do not discuss why this might be the case. The authors should address potential causes for these trends.

- **Response:** We have addressed these points in the Figure legends that they correspond to:
 - Figure 2f (now Fig. 3f): There are two potential mechanisms for increased binding of LIF to gp130: (1) D25 binding locks LIF in a favorable conformation for gp130 binding or (2) being bivalent, a complex formed of YSD-gp130-LIF-D25-LIF is possible. This would double the amount of LIF bound, which would readily explain the ~2 fold increase in binding. This is also one rationale for why One-Arm eLIFR-Fc is a better apparent gp130 competitor than bivalent eLIFR-Fc. Without the second LIFR fragment, the only complex that can be formed is YSD-gp130-LIF-One Arm eLIFR-Fc. Thus, the inhibition signal observed with eLIFR-Fc is likely underestimated.
 - *Text in Fig. 3f legend:* “**(f)** Both hLIFR-Fc and eLIFR-Fc (bivalent and one-arm) compete LIF away from WT gp130, but the D25 mAb does not and appears to increase binding, perhaps due to complex stabilization or more avid LIF binding.”
 - Figure 4a (now Supplementary Fig. 6d): The likely cause is mild agonism. It should be noted that the signal from both CTF-1 and CLCF-1 is very weak in HeLa cells (see Supplementary Fig. 6c), so a small increase in signal can look larger than it is when normalized, as is the case here. Further, CLCF-1 specifically is rather unstable, meaning that in this context it might be aided by soluble LIFR, which binds weakly, in forming a complex on the surface of cells, allowing for cell-surface LIFR to replace soluble LIFR and drive signaling. Given that LIFR is at such a high concentration, we doubt that this would be able to occur in vivo, and think that this is more of an artifact of this particular assay, and maintain the point that LIFR-Fc does not inhibit signaling for any of these cytokines.

- *Text in Supp. Fig. 6d legend: “(d) High concentrations of eLIFR-Fc or hLIFR-Fc minimally inhibit or possibly promote mild agonism of CTF-1 or CLCF-1 signaling in HeLa STAT3 cells as compared to LIF.”*

3. In Figure 1G, eLIFR demonstrates greatly elevated LIF binding RFU at saturation than hLIFR, even though expression of the two seems comparable in Figure 1E. Similarly, in Supplementary Figure 5C, hLIFR-Fc seems to only permit 50% inhibition of LIF signaling, even at substantial excess. The authors should discuss potential reasons for these observations.

○ **Response:** We have addressed these points in the Figure legends that they correspond to:

- Figure 1g (now Fig. 2g): Our lab almost always observes a higher maximum binding signal with higher affinity proteins in yeast-surface display assays. We interpret this as a slower off rate, which we show to be the case with eLIFR vs hLIFR in Fig. 3c. During the staining, washing, and flow steps, LIF is dissociating from hLIFR at a greater rate than eLIFR, which reduces the overall signal. Thus, in our eyes, both a leftward and upward shift of the binding curve in YSD binding assays are good indications of improved affinity.
 - *Text from Fig. 2g legend: “(g) Yeast-displayed eLIFR binds LIF-His with a higher affinity than WT hLIFR. Higher maximum binding signal from eLIFR likely indicates a slower off-rate of binding.”*
- Supplementary Figure 5c (now Supplementary Fig. 4c): It is true that hLIFR-Fc seems ineffective even at higher doses. In some ways this should be expected, as hLIFR-Fc has no competitive advantage over hLIFR on the surface of cells. In this assay, we used a 40-fold excess (20 nM LIFR-Fc vs 0.5 nM LIF). It is worth noting that in the IC₅₀ experiment (Fig. 3a), a 50% reduction in signal was not observed until a 90-fold excess (1.8 nM LIFR-Fc vs 0.02 nM LIF). These data suggest that hLIFR-Fc needs to be in excess of LIF by >100-fold to achieve robust inhibition, vastly higher than the concentration of eLIFR-Fc required (<2-fold).
 - *Text from Supp. Fig. 4c legend: “(c) eLIFR-Fc nearly completely silences LIF signaling, while hLIFR-Fc, with no competitive LIF binding advantage, is ineffective at silencing LIF signaling, even at concentrations 40-fold in excess.”*

4. In Figure 3 the LIF concentration used in each panel varies nearly three orders of magnitude, from 20 pM to 12 nM. Although the diversity of the tumor microenvironment makes such an estimate difficult, can the authors comment on what the potential in vivo concentration of LIF may be in a tumor? Along these same lines, how representative is the KP4-LIF model of LIF levels that may be seen in actual tumors? This should be discussed.

○ **Response:**

- Referring to data from a recent publication of our co-authors, Yu Shi and Tony Hunter (Shi, et. al., *Nature* 2019), LIF was measured in human PDAC tumors (see Shi, et. al. Fig. 5, Extended Data Fig. 10), at ~1 ng/mg protein in tumor lysate. While pancreatic lesions can vary in size, a back of the envelope calculation gives that for an 8 g tumor (~2 cm³) in which ~2% of the mass is derived from protein, LIF can be expected at a concentration of ~1 pM in this 8 mL lesion. As mentioned, this is likely an under-estimate and could vary widely by tumor, but demonstrates that testing for efficacy in the pM – nM range is reasonable.
- We reference this calculation in the methods section describing the HeLa assay, but as it is a rough calculation, do not want to put too much stock in it:
 - *Text from Materials and Methods (Lines 691 – 695): “Proteins were added to serum-starved HeLa cells for 4 h at 37 °C. For IC₅₀ analysis, eLIFR-Fc or hLIFR-Fc were*

serially diluted (eLIFR: 30 nM to 1.5 pM; hLIFR: 300 nM to 15 pM) in HeLa assay medium with a constant concentration of 20 pM LIF, reflective of the possible concentration of LIF in the pancreatic cancer microenvironment.³

- We measured LIF levels by ultra-sensitive ELISA for the KP4 tumors extracted during the study. These data are not in the revised submission itself, but to address this reviewer question, we found that on average in PBS treated KP4 tumors, LIF was present at a concentration of ~75 pg/mg of protein lysate. This concentration is lower than that estimated for patient samples, but given the relatively high level of LIF secretion observed in these cells (see above, and data repeated below), this is likely an underestimation of LIF levels, as LIF is a soluble factor, and when secreted will be diluted in the surrounding serum, and hence not effectively captured in tumor lysates. As we have found with all of the traps engineered in our lab, it is not important to just inhibit the factor from the tumor, but also the serum.

[LIF] measured in tumors treated with PBS:

[LIF] secreted into conditioned media by KP4 cells in vitro:

5. In Supplementary Figure 6H and I, 25 nM hLIFR/eLIFR is used for KP4 spheroid experiments, whereas 20-fold more (500 nM) is used for PANC1 experiments. Why the dramatic difference in concentrations?

Response: We did test lower concentrations of LIFR-Fc in PANC1, and eLIFR-Fc did somewhat reduce sphere formation at 25 nM and 150 nM (as opposed to hLIFR-Fc, which was ineffective), but it was not nearly as potent at lower concentrations. We have now included all concentrations in Fig. 5i:

Supplementary Fig. 5(i) Quantification of total sphere area in PANC1 cells after 2 weeks of growth, treated with 25 nM, 150 nM, or 500 nM eLIFR-Fc or hLIFR-Fc. [LIF] = 12 nM. ****** $P = 0.0025$, ******* $P = 0.0006$ compared to “plus LIF” or between LIFR-Fc treatments, as indicated, by two-tailed unpaired Student’s t -test. Data are the mean total sphere area \pm SD ($n = 6$).

- We found that PANC1 cells form spheres much less robustly than KP4 cells as evidenced by the small dynamic range. These cells are hence not as optimal for use in this assay, thus are presented only as a secondary cell line.

6. Overall, the Discussion is lacking in detail regarding trends seen in the in vivo models. For example, in several instances the authors observe bimodal distributions for eLIFR-Fc-treated mice in their in vivo models. What could be driving the differences in responses from tumor-to-tumor? A powerful additional experiment would be to perform immunohistochemistry or other means of quantifying LIF in excised tumors to see if intratumoral LIF levels, normalized to tumor size, correlate with response. Perhaps this is driving the bimodal distribution? Along these same lines, in Supplementary Figure 8F, the D25 combination treatment did not produce a bimodal effect in tumor volume. What might cause this difference between eLIFR-Fc and D25? This should be discussed.

- **Response:** As discussed in the response to Reviewer 1, we agree that the seemingly bimodal distribution observed in the KP4-LIF xenograft model is unusual. Given that this trend was also apparent in the PBS-treated group, it appears likely to be an effect driven by the cell line, and not treatment. KP4-LIF is a heterogenous cell line (not derived from a single clonal population after transduction).
- As discussed above, due to the unusual nature of this xenograft model, and our success in optimizing the KP4 xenograft model, we are replacing the KP4-LIF data with the new KP4 data in the revised manuscript.
- We analyzed LIF levels in tumor lysates from the KP4 WT study by ultra-sensitive ELISA, summarized below. There was no correlation observed between tumor size and LIF expression in the PBS treated group, despite a range of LIF protein levels. There was a moderate correlation between LIF levels and tumor size in the eLIFR-Fc treated group ($R^2=0.70$), however, not enough data to draw a strong conclusion.

LIF levels in KP4 WT tumor lysates. (a) LIF levels quantified by ultrasensitive ELISA from lysates of KP4 WT tumors treated with PBS or eLIFR-Fc. Data are the mean \pm SD ($n = 7$). **(b)** Correlation of LIF levels quantified by ultrasensitive ELISA from lysates of KP4 tumors with excised tumor weight. Treatment groups are shown as different colors (PBS: black; Purple: eLIFR-Fc). **(c)** Correlation between LIF levels and excised tumor weight in individual treatment groups, as described in **b**.

7. It is impressive that mLIFR binds human LIF with such high affinity. The authors mention that three of the mutations in eLIFR are found in mLIFR, but what might be other reasons for the high affinity binding observed with mLIFR? This should be discussed.

- **Response:** This is a challenging question, and one that we have explored with computational modeling, but which does not have a readily apparent answer. In all, it seems to be the contributions of many residues and regions, though it is beyond the scope of this work to profile them.
- We allude to one possible explanation/solution through identifying binding interactions present in other domains of LIFR besides the Ig-like domain in the Discussion:
 - *Text from Discussion (Lines 348 – 351):* “Further improvements in affinity and species cross-reactivity may thus be possible through insertion of other regions of mLIFR that contact LIF (such as the CBM II) into the eLIFR scaffold, however, additional changes may result in increased immunogenicity.”

Minor Comments:

1. On page 11, line 236, there is a typo. “Even at LIF concentrations of only ten-fold excess...” should read “Even at LIFR concentrations of only ten-fold excess...”.

- **Response:** Good catch! This change has been made.

2. In Supplementary Figure 4B the authors clearly spliced images of different gels together. Even if it is just the molecular weight marker, clear delineation between different gels (e.g. a black line) should be shown to demonstrate that different gels are b

- **Response:** Thank you! It was not our intent to have these gels seem spliced together – lanes are placed together for clarity. We have put lines between the gels to indicate that they have been spliced together to avoid confusion that the lanes were run side by side. The legend now makes clear that the lanes were taken from the same gel and have now been placed side-by-side for clarity with lines indicating lanes that were not adjacent on the original gel.